# LoRA-S: An Efficient Low Rank Adaptation scheme via Sylvester equation

**Jinyang Zheng**[1]**, Tong Wu**[1]

[1]Department of Mathematics, The Hong Kong University of Science and Technology,
Hong Kong, China
{jzhengbp,twubi}@connect.ust.hk

## Abstract

Numerous studies on low-rank adaptation (LoRA) emerged in recent years, with the aim of accelerating the convergence of the LoRA framework. In this paper, we leverage the horizontal lift theory from differential geometry to establish the general iteration scheme on the quotient manifold $\mathbb{R}_*^{m \times r} \times \mathbb{R}_*^{n \times r} / \sim$. By endowing the LoRA framework with Riemannian quotient geometries, our theory not only guarantees efficient feature learning but also bridges the LoRA algorithms and the pre-training algorithms for large models. Furthermore, we theoretically analyze the role of the weight decay matrix $\epsilon_{decay} I$ in efficient feature learning and then replace it with the Sylvester matrix $K$, indicating that the theory helps remove an important hyperparameter while generating accurate and computationally efficient optimizers. Based on the general scheme, we propose two efficient LoRA optimizers with runtime analysis, Adam-Sylvester (AdamS) and LRACS, then conduct experiments on the transformer-based networks. The results demonstrate evident improvements over existing optimizers. The codes are available at https://gitee.com/sanjin998/lora_s.

## 1 Introduction

Low-Rank Adaptation (LoRA) (Hu et al., 2022) is a widely used parameter efficient fine-tune (PEFT) method for training large language models. By freezing the pre-trained weights $W \in \mathbb{R}^{n \times m}$ and adding a trainable low-rank matrix $X = MN^\top \in \mathbb{R}^{n \times m}$ to each layer where $W$ appears, LoRA can achieve: (1) reduced memory requirements for fine-tuning, and (2) avoidance of overfitting with relatively little training data. Let $f_{\mathcal{L}}(W)$ denote for the original loss function and $\mathcal{L}(M, N)$ denote the factorial version of the loss function. In each layer, training LoRA is equivalent to minimize the loss function in the space $\mathbb{R}^{m \times r} \times \mathbb{R}^{n \times r}$:

$$\min_{(M,N) \in \mathbb{R}^{m \times r} \times \mathbb{R}^{n \times r}} \mathcal{L}(M, N) = f_{\mathcal{L}}(W + X), \quad X = MN^T$$

Recent research explores numerous variations and improvements in the LoRA algorithm (Zhang et al., 2023; Valipour et al., 2022; Hayou et al., 2024; Yen et al., 2025). Among them, **efficient feature learning** (EFL) (Hayou et al., 2024) is particularly influential. Since it is an ease-to-verify mathematical property that ensures that both factors $M, N$ are updated efficiently. Unfortunately, conventional methods cannot achieve efficient feature learning. To achieve this property, LoRA+ (Hayou et al., 2024) employs different learning rates for the two factors, which needs more fine-tuning, and LoRA-Rite (Yen et al., 2025) designs a novel and sophisticate optimizer to achieve the property. These methods either require multiple rounds of fine-tuning, or their implementations are overly complex, i.e. redesign the preconditioner, and hard to generalize.

To address this challenge, we introduce an equivalence relation $\sim$, see (Definition 1), and propose our general scheme (Algorithm 1) by applying horizontal lift theory (Absil et al., 2014) on the quotient manifold $\mathbb{R}_*^{m \times r} \times \mathbb{R}_*^{n \times r} / \sim$. Through theoretical analysis (Theorem 1), optimizers conforming to our general scheme are guaranteed to achieve transformation invariance (Yen et al., 2025), thus ensuring efficient feature learning (EFL). The general scheme is highly flexible and scalable. It can endow almost all conventional optimizers with transformation invariance and is capable of maintaining EFL even after any preconditioning is applied, which is not supported by former Riemannian

optimization methods such as RGD (Bonnabel, 2013) and ScaledAdam (Zhang & Pilanci, 2024). By introducing the state-of-the-art preconditioning to mitigate the condition number of the Hessian matrix in LLMs, our new EFL approach has certain advantages over existing efficient optimizers, i.e. LoRA-Rite (Yen et al., 2025) in all metrics and significantly exceeds the simple EFL methods, i.e. Scaled GD (Tong et al., 2021) and Quotient GD (Mishra et al., 2014), used as the baseline.

Fine-tuning hyperparameters is also a crucial part of LoRA training. Most conventional optimizers require simultaneous fine-tuning of both the learning rate and the weight decay parameter, thus searching for the optimal weight-decay parameter $\lambda^*$ can be very time-consuming. However, we find that the weight-decay hyperparameter is unnecessary in our framework. Its benefit to EFL is far less significant than that of our proposed Sylvester equation. Both theoretical and empirical results show that the effect of efficient feature learning exceeds the effect of $L_2$-regularization. Considering that the Sylvester decay matrix $K_t$ can be computed directly without tuning hyperparameter, our weight-decay-free methods can practically save a substantial amount of time.

The contribution of this paper can be summarized below:

- We propose LoRA-S, an efficient low rank adaption scheme via Sylvester equation which can convert optimizers with any preconditionings to transformation-invariant optimizers. By proposing a formal math definition of efficient feature learning which is more strict than transformation invariance, we first provide a perspective to understand EFL methods with manifold optimization.
- We also study the role of weight-decay hyperparameter in efficient feature learning. In our scheme, we find removing the weight decay parameter $\lambda$ and replacing it with the Sylvester decay matrix $K_t$ can achieve more efficient learning, both theoretically and empirically.
- We lift the two conventional optimizers, Adam (Kingma & Ba, 2014) and RACS (Gong et al., 2025), proposing AdamS and LRACS. Empirically, our proposed AdamS method achieves an average CLIP score of 32.64 on the Mix-of-Show model—an improvement of 9% over the widely used Adam optimizer (29.86) and 3% over LoRA-Rite (31.90), the current state-of-the-art EFL method.

## 2 PRELIMINARIES

### 2.1 DIFFERENTIAL GEOMETRY

In this section, we briefly introduce concepts from differential geometry and explain how Low-Rank Adaptation (LoRA) (Hu et al., 2022) framework fits in manifold optimization. Recall that $X = MN^{\mathrm{T}}$ belongs to the manifold $\mathcal{M}(r, m \times n)$ which denotes the set of all rank-$r$ matrices of size $m \times n$,

$$\mathcal{M}(r, m \times n) = \left\{ X \in \mathbb{R}^{m \times n} : \mathrm{rank}(X) = r \right\}.$$

Let $\mathbb{R}_*^{m \times r} = \left\{ X \in \mathbb{R}^{m \times r} : \mathrm{rank}(X) = r \right\}$ denotes the set of all full-rank $m \times r$ matrices. The product space $\mathbb{R}_*^{m \times r} \times \mathbb{R}_*^{n \times r}$ forms a manifold. Compared to the non-linear $\mathcal{M}(r, m \times n)$, $\mathbb{R}_*^{m \times r} \times \mathbb{R}_*^{n \times r}$ is the linear space $\mathbb{R}^{m \times r} \times \mathbb{R}^{n \times r}$ with a nowhere dense set excerpted. Therefore we usually optimize LoRA in the ambient space $\mathbb{R}^{m \times r} \times \mathbb{R}^{n \times r}$ in practice. Meanwhile, the tangent space of manifold $\mathbb{R}_*^{m \times r} \times \mathbb{R}_*^{n \times r}$ at $(M, N)$ and the tangent space of manifold $\mathcal{M}(r, m \times n)$ at $MN^T$ (Séguin et al., 2024) are:

$$\mathrm{T}_{(M,N)}\mathbb{R}_*^{m \times r} \times \mathbb{R}_*^{n \times r} = \mathbb{R}^{m \times r} \times \mathbb{R}^{n \times r}, \tag{1}$$

$$\mathrm{T}_{(MN^T)}\mathcal{M}(r, m \times n) = \left\{ T \in \mathbb{R}^{m \times n} : T = MSN^\top + M_{\mathrm{p}}N^\top + MN_{\mathrm{p}}^\top \right\}, \tag{2}$$

where $S \in \mathbb{R}^{r \times r}$, $M_{\mathrm{p}} \in \mathbb{R}^{m \times r}$, and $N_{\mathrm{p}} \in \mathbb{R}^{n \times r}$ satisfy $M^\top M_{\mathrm{p}} = 0$ and $N^\top N_{\mathrm{p}} = 0$. For convenience, we denote $\mathrm{T}_{(M,N)}\mathbb{R}_*^{m \times r} \times \mathbb{R}_*^{n \times r}$ as $\mathrm{T}_{(M,N)}$, and $\mathrm{T}_{(MN^T)}\mathcal{M}(r, m \times n)$ as $\mathrm{T}_{(MN^T)}$. Notably, a map $\pi$ can be defined to transport tangent vectors between these two tangent spaces via horizontal lift theory.

### 2.2 HORIZONTAL LIFT THEORY

The horizontal lift theory of fixed-rank matrices was firstly studied in (Absil et al., 2014). It is said that, there exists a Riemannian submersion $\pi$ from the product space of two full-rank matrices

$\mathbb{R}_*^{m \times r} \times \mathbb{R}_*^{n \times r}$ to fixed-rank matrices manifold $\mathcal{M}(r, m \times n)$:

$$\pi : \mathbb{R}_*^{m \times r} \times \mathbb{R}_*^{n \times r} \to \mathcal{M}(r, m \times n) : (M, N) \mapsto M N^{\mathrm{T}}.$$

The degree of freedom (d.f.) of $\mathcal{M}(r, m \times n)$ is $mr + nr - r^2$ while the degree of freedom of $\mathbb{R}_*^{m \times r} \times \mathbb{R}_*^{n \times r}$ is $mr + nr$. The Riemannian submersion $\pi$ is a projection. Furthermore, given a point $M N^T \in \mathcal{M}(r, m \times n)$ we can also write down its preimage:

$$\pi^{-1} \left( M N^{\mathrm{T}} \right) = \left\{ \left( M R, N R^{-\mathrm{T}} \right) : R \in \mathrm{GL}(r) \right\},$$

where

$$\mathrm{GL}(r) = \left\{ R \in \mathbb{R}^{r \times r} : \det(R) \neq 0 \right\}$$

denotes the general linear group of degree $r$. From (Absil et al., 2008, Proposition 3.3.3), we know that fibers $\pi^{-1}(M N^T)$ are $r^2$-dimensional submanifolds of $\mathbb{R}_*^{m \times r} \times \mathbb{R}_*^{n \times r}$. The redundancy leads most conventional methods to become transformation-variant (Yen et al., 2025) and miss efficient feature learning. To eliminate redundant dimensions, (Absil et al., 2014) defines the equivalence relation $\sim$ on $\mathbb{R}_*^{m \times r} \times \mathbb{R}_*^{n \times r}$:

**Definition 1** (Equivalence relation).

$$(M_a, N_a) \sim (M_b, N_b) \text{ if and only if } \pi(M_a, N_a) = \pi(M_b, N_b)$$

In this way, $\mathbb{R}_*^{m \times r} \times \mathbb{R}_*^{n \times r} / \sim$ is a quotient manifold diffeomorphic to $\mathcal{M}(r, m \times n)$. Let $(\dot{M}, \dot{N}) \in \mathbb{R}^{m \times r} \times \mathbb{R}^{n \times r}$ be the update pair, and also the tangent vectors, at $(M, N)$. Then the derivative of $\pi$ will give that:

$$D\pi(M, N) : \mathrm{T}_{(M,N)} \to \mathrm{T}_{(M N^T)} : (\dot{M}, \dot{N}) \mapsto \dot{M} N^{\mathrm{T}} + M \dot{N}^{\mathrm{T}}.$$

The horizontal lift theory (Absil et al., 2014) builds the relation of the tangent vectors between $\mathrm{T}_{(M,N)}$ and $\mathrm{T}_{(M N^T)}$. It is said that, given a tangent vector $\dot{X}_{M N^{\mathrm{T}}} \in \mathrm{T}_{M N^{\mathrm{T}}}$, there is one and only one lift vector:

$$\dot{X}_{\uparrow(M,N)} \in \mathcal{H}_{(M,N)} \quad \text{such that} \quad D\pi(M, N) \left[ \dot{X}_{\uparrow(M,N)} \right] = \dot{X}_{M N^{\mathrm{T}}},$$

where $D\pi(X)[\dot{X}]$ denotes the differential of $\pi$ at $X$ applied to $\dot{X}$, and $\bar{g}$ is a specific metric on $\mathbb{R}_*^{m \times r} \times \mathbb{R}_*^{n \times r}$. Equipped with the metric $\bar{g}$, the tangent space $\mathrm{T}_{(M,N)}$ can be decomposed as a direct sum of the kernel of $D\pi(M, N)$ and its orthogonal complement, called the horizontal space $\mathcal{H}_{(M,N)} \subseteq \mathrm{T}_{(M,N)} = \mathbb{R}^{m \times r} \times \mathbb{R}^{n \times r}$:

$$\mathcal{H}_{(M,N)} = \left\{ (\dot{M}, \dot{N}) \in \mathbb{R}^{m \times r} \times \mathbb{R}^{n \times r} : \bar{g}_{(M,N)} \left( (\dot{M}, \dot{N}), \left( M \dot{R}, -N \dot{R}^{\mathrm{T}} \right) \right) = 0, \forall \dot{R} \in \mathbb{R}^{r \times r} \right\}. \tag{3}$$

Note that the $r^2$ linear constraints in the definition successfully remove the redundant dimensions, i.e., $\dim(\ker(D\pi)) = r^2$ and as long as the redundancy is removed, the transformation dependence will be removed. Since d.f. of $\mathrm{T}_{(M N^T)}$ is $mr + nr - r^2$ and d.f. of $\mathrm{T}_{(M,N)}$ is $mr + nr$, the vector $\dot{X}_{\uparrow(M,N)}$ is termed the horizontal lift of $\dot{X}_{M N^T}$ at $(M, N)$. And we call $\mathrm{T}_{(M N^T)}$ the matrix tangent space and $\mathrm{T}_{(M,N)}$ the lifting space.

## 3 Efficient LoRA iteration scheme

In this section, we first give a formal mathematical definition of **efficient feature learning** (EFL) then show how the theory in Section 2 can help build a general iteration scheme that guarantees efficient feature learning.

### 3.1 Efficient Feature Learning

The concept of efficient feature learning (Hayou et al., 2024) describes the asymptotic training behavior of LoRA as the network width grows. Efficient feature learning requires that both $\dot{X}_{\uparrow \mathrm{M}(M_t, N_t)} N_t^{\mathrm{T}} w$ and $M_t \dot{X}_{\uparrow \mathrm{N}(M_t, N_t)}^{\mathrm{T}} w$ are of magnitude $\Theta(n^0) = \Theta(1)$ where $n$ is the width of network and $w$ is the input embedding. The explosion of the gradient occurs when the magnitude reaches $\Theta(n^\alpha)$ ($\alpha > 1$) and the gradient vanishes when the magnitude is below $\Theta(n^\alpha)$ ($\alpha < 1$). In other words, keeping both $\delta_t^1 = \|\dot{M} N^{\mathrm{T}}\|$ and $\delta_t^2 = \|M \dot{N}^{\mathrm{T}}\|$ at same magnitude means that both $M$ and $N$ parameter updates significantly contribute to the change in $\mathcal{L}(M, N)$. Based on these properties, we propose our formal mathematical definition of efficient feature learning with LoRA:

**Definition 2** (Efficient feature learning with LoRA). Suppose $(M_1, N_1)$, $(M_2, N_2)$ are two pairs of LoRA factors that represent the same weight, that is, $\delta W = M_1 N_1^{\mathrm{T}} = M_2 N_2^{\mathrm{T}}$. Then w.l.o.g. we can assume $M_2 = M_1 R$, $N_2 = N_1 R^{-\mathrm{T}}$, where $R \in \mathrm{GL}(r)$. We say an update scheme satisfy efficient feature learning if the updates $(\dot{M}_1, \dot{N}_1)$ at $(M_1, N_1)$ and the updates $(\dot{M}_2, \dot{N}_2)$ at $(M_2, N_2)$ shares the same magnitude w.r.t. a specific metric $\bar{g}$:

$$\bar{g}_{(M_1, N_1)}\left((\dot{M}_1, \dot{N}_1), (\dot{M}_1, \dot{N}_1)\right) = \bar{g}_{(M_2, N_2)}\left((\dot{M}_2, \dot{N}_2), (\dot{M}_2, \dot{N}_2)\right), \tag{4}$$

and the updates should satisfy:

$$\dot{M}_2 = \dot{M}_1 R, \quad \dot{N}_2 = \dot{N}_1 R^{-\mathrm{T}} \tag{5}$$

The definition (2) ensures that the magnitudes of updates, $(\dot{M}_1, \dot{N}_1)$ and $(\dot{M}_2, \dot{N}_2)$, remain the same. From the perspective of manifold optimization, equation (4) is the necessary and sufficient condition for the existence of the induced metric $g_{MN^{\mathrm{T}}}$ on tangent space $\mathrm{T}_{(MN^{\mathrm{T}})}$. In addition, note that equation (5) also indicates the transformation invariance property which means that our definition has close relation with former studies:

**Proposition 1.** The transformation invariance scheme in LoRA-Rite (Yen et al., 2025) indicates efficient feature learning (Definition 2) with metric:

$$\bar{g}_{(M, N)}\left((\dot{M}, \dot{N}), (\dot{M}, \dot{N})\right) = \mathrm{trace}(N^{\mathrm{T}} N \dot{M}^{\mathrm{T}} \dot{M} + M^{\mathrm{T}} M \dot{N}^{\mathrm{T}} \dot{N}) \tag{6}$$

The proof of this proposition is given in Appendix (A.4).

## 3.2 DEFINITION OF THE ITERATION SCHEME

In this section, we choose the metric used in Grassmannian quotient manifold (Absil et al., 2007, Example 3.6.4) that meets efficient feature learning condition (2) and apply horizontal lift theory to calculate the lifted vectors $(\dot{X}_{\mathrm{M}(M,N)}, \dot{X}_{\mathrm{N}(M,N)})$ as updates. Our metric induces update rules with richer structure and interpretability than the metric (13), which will be shown in (A.18.9). The metric $\bar{g}$ on $\mathbb{R}_*^{m \times r} \times \mathbb{R}_*^{n \times r}$ is:

$$\bar{g}_{(M, N)}((\dot{M}, \dot{N}), (\check{M}, \check{N})) := \mathrm{trace}\left(\left(M^{\mathrm{T}} M\right)^{-1} \dot{M}^{\mathrm{T}} \check{M} + \left(N^{\mathrm{T}} N\right)^{-1} \dot{N}^{\mathrm{T}} \check{N}\right) \tag{7}$$

$$= g_{MN^{\mathrm{T}}}\left(\dot{X}_{MN^{\mathrm{T}}}, \check{X}_{MN^{\mathrm{T}}}\right) \tag{8}$$

where $(\dot{M}, \dot{N})$ and $(\check{M}, \check{N})$ are tangent vectors. Then let $X = MN^{\mathrm{T}}$ and let $\dot{X}_{MN^{\mathrm{T}}}$ belong to $\mathrm{T}_{(MN^{\mathrm{T}})}$. The horizontal lift $\dot{X}_{\uparrow(M,N)} = \left(\dot{X}_{\uparrow\mathrm{M}(M,N)}, \dot{X}_{\uparrow\mathrm{N}(M,N)}\right)$ derived from (3) is:

$$\dot{X}_{\uparrow\mathrm{M}(M,N)} = \left(\dot{X}_{MN^{\mathrm{T}}} N - MN^{\mathrm{T}} N K\right) \left(N^{\mathrm{T}} N\right)^{-1}$$

$$\dot{X}_{\uparrow\mathrm{N}(M,N)} = \left(\dot{X}_{MN^{\mathrm{T}}}^{\mathrm{T}} M - NM^{\mathrm{T}} M K^{\mathrm{T}}\right) \left(M^{\mathrm{T}} M\right)^{-1},$$

where $K$ is the Sylvester matrix such that

$$M^{\mathrm{T}} \dot{X}_{MN^{\mathrm{T}}} N = M^{\mathrm{T}} M N^{\mathrm{T}} N K + K M^{\mathrm{T}} M N^{\mathrm{T}} N. \tag{9}$$

The equation is called the Sylvester equation, and in our case there exists a unique solution matrix $K$. We conclude the general iteration scheme in Algorithm (1). From the analysis in the previous section, we know that:

**Theorem 1.** Any optimizer following the general scheme (Algorithm 1) achieves efficient feature learning (Definition 2) and will become transformation invariant.

The proof is given in Appendix A.5. While the conventional optimizers being not transformation invariant, researchers can transform any conventional optimizers to transformation invariant optimizers now by applying the general scheme. See Appendix A.18 for a unified and comprehensive analysis of conventional optimizers under our general scheme.

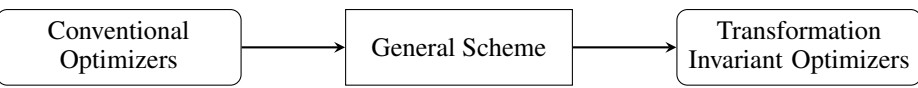

---

**Algorithm 1** General Scheme

---

1: **for** $t = 1 \dots T$ **do**
2:     Assign the descent vector $\dot{X}_{M_t N_t^{\mathrm{T}}}$
3:     Solve $K_t$ from the Sylvester equation (9)
4:     Compute the lift vector $\dot{X}_{\uparrow \mathrm{M}(M_t, N_t)} = \left( \dot{X}_{M_t N_t^{\mathrm{T}}} N_t - M_t N_t^{\mathrm{T}} N_t K_t \right) \left( N_t^{\mathrm{T}} N_t \right)^{-1}$
5:     Compute the lift vector $\dot{X}_{\uparrow \mathrm{N}(M_t, N_t)} = \left( \dot{X}_{M_t N_t^{\mathrm{T}}}^{\mathrm{T}} M_t - N_t M_t^{\mathrm{T}} M_t K_t^{\mathrm{T}} \right) \left( M_t^{\mathrm{T}} M_t \right)^{-1}$
6:     Update model parameters for $(M_{t+1}, N_{t+1}) \leftarrow (M_t + \dot{X}_{\uparrow \mathrm{M}(M_t, N_t)}, N_t + \dot{X}_{\uparrow \mathrm{N}(M_t, N_t)})$
7: **end for**

---

### 3.3 REMOVING THE WEIGHT DECAY PARAMETER

In machine learning, weight decay (Loshchilov & Hutter, 2017) is a critical regularization technique primarily used to prevent overfitting of the model. It is typically implemented by adding a penalty term proportional to the square of the model weights ($\mathrm{L}_2$ norm) to the loss function. The $\mathrm{L}_2$ regularizer $\epsilon_{decay}$ is often set to $\eta \times \lambda$, where $\eta$ is the learning rate and $\lambda$ is the weight decay value.

$$\epsilon_{decay} = \eta \times \lambda \tag{10}$$

To maintain training stability and consistent regularization effects, it is common to adjust the weight decay coefficient $\lambda$ proportionally when modifying the learning rate $\eta$, which will cost a lot of time to find the overall best weight decay value $\lambda^*$ in practice. Practically, $\lambda^*$ is non-zero in most non-EFL LoRA optimizers, i.e. 0.01 in AdamW and ScaledAdamW (Zhang & Pilanci, 2024), while being set to 0 in EFL LoRA optimizers, i.e. LoRA+ (Hayou et al., 2024) and LoRA-Rite (Yen et al., 2025). In this section, we will theoretically explain this phenomenon and why this parameter can be removed in our scheme.

$$\dot{X}_{\uparrow \mathrm{M}(M_t, N_t)} = \underbrace{\dot{X}_{M_t N_t^{\mathrm{T}}} N_t \left( N_t^{\mathrm{T}} N_t \right)^{-1}}_{\text{gradient term (GT)}} - \underbrace{M_t N_t^{\mathrm{T}} N_t K_t \left( N_t^{\mathrm{T}} N_t \right)^{-1}}_{\text{decay term (DT)}}$$

First, the lifting vectors $\dot{X}_{\uparrow \mathrm{M}(M_t, N_t)}$ and $\dot{X}_{\uparrow \mathrm{N}(M_t, N_t)}$ consist of two terms. The first term is called the "gradient term" since it directly contains the lifted gradient or preconditioned gradient vector. The second term is called the "decay term" since it contains $-M_t$. The weight decay parameter in the LoRA framework can be viewed as an approximation to the "decay term". Scaling the lifted vector will scale $K_t$ and "decay term" at the same time, i.e.,

$$M_t^{\mathrm{T}}(\eta \dot{X}_{M_t N_t^{\mathrm{T}}}) N_t = M_t^{\mathrm{T}} M_t N_t^{\mathrm{T}} N_t (\eta K_t) + (\eta K_t) M_t^{\mathrm{T}} M_t N_t^{\mathrm{T}} N_t.$$

In this sense, the magnitude of the Sylvester decay matrix $\eta K_t$ is determined by two decoupled parameters, the hyperparameter learning rate $\eta$ and the solution matrix $K_t$, just like the situation of AdamW and SGDW (Loshchilov & Hutter, 2017) in (10). Conventional methods typically do not solve the Sylvester equation; instead, they use an approximate matrix $K_\epsilon$ to estimate the Sylvester matrix $K$, taking $M_t$ for example,

$$-\epsilon_{decay} M_t = -M_t(N_t^{\mathrm{T}} N_t) K_\epsilon (N_t^{\mathrm{T}} N_t)^{-1} , \; K_\epsilon = \epsilon_{decay} I$$

Of course, such estimation will result in computational errors in the Sylvester equation (9). The errors $E_{K_t}$ can be bounded by the norm of $\epsilon_{\text{decay}} M_t N_t^{\mathrm{T}}$ and $\dot{X}_{M_t N_t^{\mathrm{T}}}$:

$$E_{K_t} = M_t^{\mathrm{T}} \dot{X}_{M_t N_t^{\mathrm{T}}} N_t - M_t^{\mathrm{T}} M_t N_t^{\mathrm{T}} N_t K_\epsilon - K_\epsilon M_t^{\mathrm{T}} M_t N_t^{\mathrm{T}} N_t$$

$$= M_t^{\mathrm{T}} (\dot{X}_{M_t N_t^{\mathrm{T}}} - 2\epsilon_{\text{decay}} M_t N_t^{\mathrm{T}}) N_t$$

$$\text{(relative error)} \quad R_{K_t} = ||\dot{X}_{M_t N_t^{\mathrm{T}}} - 2\epsilon_{\text{decay}} M_t N_t^{\mathrm{T}}|| \geq \frac{||E_{K_t}||}{||M_t|| \times ||N_t||}$$

Therefore, fine-tuning the weight decay parameter is actually minimizing the computational errors of (9) in conventional methods. Less computational errors means more efficient feature learning, while the efficient optimizers (Hayou et al., 2024) (Yen et al., 2025) may not need to apply weight decay since they already achieve efficient feature learning. Empirically, we computed the norm of

relative error in Adam optimizer and note that the error decreases by iterations. And weight-decay param $\lambda$ that yields lower relative error tends to achieve lower EMA loss in experiments. It gives us intuition to eliminate the relative error through Sylvester equation. See Fig. (3). A similar result about SGD is given in Appendix (A.6). In addition to these experiments, we also provide a practical intuition for the weight-decay matrix vs. sylvester matrix from the perspectives of the update norm and preconditioning in Appendix (A.7).

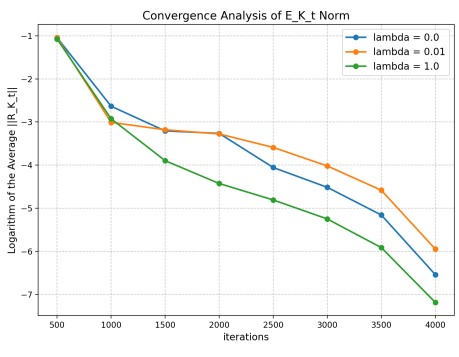

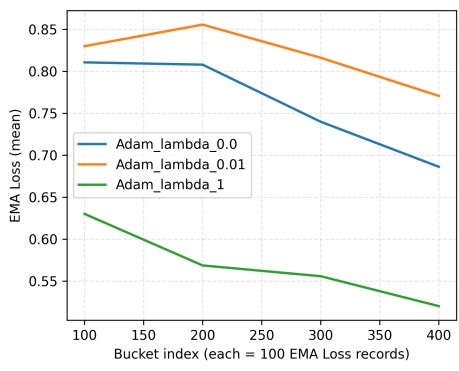

(a) Relative error of Adam for $\lambda = 0, 0.01, 1$

(b) EMA Loss of Adam for $\lambda = 0, 0.01, 1$

Figure 1: Weight decay $\lambda$ that yields lower relative error tends to achieve lower EMA loss.

**Effect of regularization.** Although, according to our experiment (3), the logarithm of the relative error $R_{K_t}$ is mainly between $1e^{-4}$ and $1e^{-12}$, which means that adding the appropriate weight decay matrix can be viewed as a good approximation to efficient feature learning. We still need to verify whether adding $L_2$ regularization to the loss function can improve performance because adding regularization to $K_t$ contradicts efficient feature learning. We perform relevant experiments and find that the benefits of achieving accurate efficient feature learning outweigh the benefits of applying $L_2$ regularization. See Appendix (A.11).

**Cost of removing weight-decay.** In fact, computing the Sylvester decay matrix is not cost-free. According to the Bartels–Stewart algorithm (Bartels & Stewart, 1972), it requires $O(r^3)$ flops to compute the matrix $K_t$ at each timestep, meaning that the computation of $K_t$ is fast only when $r \ll \min(m, n)$. However, our method can avoid repeated experiments to search for $\lambda^*$, which saves the experimental time significantly. To validate our approach, we conducted experiments with our weight-decay-free methods, which turned out to achieve even better performance than conventional optimizers.

## 4 OUR PROPOSED METHODS

In this section, we gain intuition from optimizers proven effective in pre-training and present the corresponding lifted algorithms. These lifted algorithms far surpass conventional LoRA optimizers and achieve more efficient feature learning in transformer-based networks.

### 4.1 PRECONDITIONING ON THE MATRIX TANGENT SPACE

Our intuition comes from efficient optimizers for large language models (LLMs) with low-memory requirements and fast convergence. According to the recent research (Gong et al., 2025), state-of-the-art optimizers can be viewed as solutions to Fisher Information Matrix (FIM) approximation (under the Frobenius norm). Suppose that the approximate FIM $\tilde{\boldsymbol{F}} \in \{\text{Diag}_v(\boldsymbol{v}); v_i > 0\}$ and $\boldsymbol{F}$ is the empirical FIM. Then minimizing the objective $\min_{\tilde{\boldsymbol{F}}} \|\tilde{\boldsymbol{F}} - \boldsymbol{F}\|_F^2$ yields the following analytic solution: $\tilde{\boldsymbol{F}}^* = \text{Diag}\left(\mathbb{E}\left[g^2\right]\right)$, where $g^2$ indicates the element-wise square of the gradient $g$. This diagonal approximation $\tilde{\boldsymbol{F}}^*$ gives the form of Adam:

$$\text{vec}(\tilde{X}_{M_t N_t^{\mathrm{T}}}) = (\tilde{\boldsymbol{F}}^*)^{-1/2}\text{vec}(\dot{X}_{M_t N_t^{\mathrm{T}}}),$$

where $\mathrm{vec}(\cdot)$ means viewing the elements of a matrix as a vector in column-major order. Applying this kind of Gauss-Newton-like methods on the matrix tangent space $\mathrm{T}_{(MN^T)}\mathcal{M}(r, m \times n)$ can guarantee efficient feature learning while most methods failed to achieve efficient feature learning by applying preconditioner to the lifting space $\mathbb{R}^{m \times r} \times \mathbb{R}^{n \times r}$.

To see this, suppose the horizontal lifts of $\tilde{X}_{M_t N_t^T}$ at $(M, N)$ and $(MR, NR^{-T})$ are $(\dot{M}, \dot{N})$ and $(\ddot{M}, \ddot{N})$ where $R \in \mathrm{GL}(r)$. Since they are vectors on the fiber $\pi^{-1}(M_t N_t^T)$, the following invariance equation always holds:

$$\ddot{M} = \dot{M}R, \quad \ddot{N} = \dot{N}R^{-T} \tag{11}$$

This property can also be seen from the analysis in A.5, And (11) leads to the transformation invariance condition (5). Since the dimension of the matrix tangent space, i.e. $\mathrm{T}_{M_t N_t^T}$, is only $mr + nr - r^2$, the solution of the Sylvester equation (9) $K$ only recovers the information of $\tilde{X}_{MN^T}$ on $\mathrm{T}_{M_t N_t^T}$. Thus, mathematically, we need to project $\tilde{X}_{MN^T}$ back to $\mathrm{T}_{M_t N_t^T}$ to avoid computational errors. We provide a detailed analysis in Appendix (A.8).

## 4.2   AdamS – Lifting Adam to horizontal space

**Assigning the descent vector.**   The descent vector exists on the matrix tangent space and it is usually set to be the projected gradient, i.e.,

$$\dot{X}_{M_t N_t^T} \leftarrow \mathrm{P}^{(e)}_{\mathrm{T}_{M_t N_t^T}}(\nabla_{M_t N_t^T} f_{\mathcal{L}}(M_t N_t)).$$

An accurate but costly way to obtain $\dot{X}_{M_t N_t^T}$ is to use hook grad. When the hook is registered correctly and called during back propagation, it will return $\nabla_{M_t N_t^T} f_{\mathcal{L}}(M_t N_t)$. Another method (Yen et al., 2025) applies QR decomposition (Gander, 1980) to $M_t, N_t$ and then uses the gradients of the two factors, $\nabla M_t$ and $\nabla N_t$, to recover $\dot{X}_{M_t N_t^T}$. Assuming $M_t = UR_{M_t}, N_t = VR_{N_t}$,

$$\dot{X}_{M_t N_t^T} \leftarrow U(R_{M_t}^{\dagger})^T \nabla N_t^T + \nabla M_t R_{N_t}^{\dagger} V^T - U(R_{M_t}^{\dagger})^T \nabla N_t^T VV^T. \tag{12}$$

We adopt the efficient though may not accurate form in (12), which works well in our experiments.

**Incorporating moments.**   From the previous section 4.1, we are about to complete the lifting version of Adam. We employ a simple accumulation of first and second moments as in AdamW (Loshchilov & Hutter, 2017). Suppose the 1$^{\mathrm{st}}$ order decay parameter is $\beta_1$ and the 2$^{\mathrm{nd}}$ order decay parameter is $\beta_2$, we have:

$$m_t \leftarrow \beta_1 m_{t-1} + (1 - \beta_1)\dot{X}_{M_t N_t^T},$$
$$v_t \leftarrow \beta_2 v_{t-1} + (1 - \beta_2)\dot{X}^2_{M_t N_t^T}. \tag{13}$$

Besides, there are different ways to incorporate moments in related methods. LoRA-Rite (Yen et al., 2025) adopts a varying basis by projecting the moments at each step $P_t = U_t^T U_{t-1}, m_t^M = m_{t-1}^M P_t^T$, which introduces mass loss $\rho_t$ and requires additional calculations for the escape mass $d_\lambda$. Meanwhile, Riemannian adaptive methods (Becigneul & Ganea, 2019) prefer to accumulate second moments under Riemannian metric (8) $g$. That is, by replacing (13) with:

$$v_t \leftarrow \beta_2 v_{t-1} + (1 - \beta_2)\dot{X}^g_{M_t N_t^T},$$

where $\dot{X}^g_{M_t N_t^T} = \frac{||\dot{X}^2_{M_t N_t^T}||_g}{||\dot{X}^2_{M_t N_t^T}||_2}\dot{X}^2_{M_t N_t^T}$ is the element-wise product under $g$. Computing $\dot{X}^g_{M_t N_t^T}$ can improve training, though it requires solving an extra Sylvester equation and incurs about 10–20% overhead. Therefore, we provide the Riemannian inner product as an optional alternative to the standard inner product (13). This completes the Adam lifting, termed AdamS (S for Sylvester), with minimal changes. The full iteration scheme is given in Algorithm 2 (Appendix A.9).

## 4.3   LRACS – Lifting RACS to horizontal space

Although AdamS (Algorithm 2) is simple and efficient, it requires relatively high memory, i.e. the storage of $m_t$ and $v_t$ consumes $2mn$ memory. We propose a skillful trick to halve the memory

consumption of AdamS as a low-memory version in Appendix (A.17). However, an easier way is to lift an efficient and low-memory method from pre-training other than Adam. Gong et al. (2025) proposed RACS algorithm for LLM pre-training. They assume that FIM $\tilde{F} \in \{S \otimes Q\}$ where $S \in \mathbb{R}^{n \times n}, Q \in \mathbb{R}^{m \times m}$ are positive diagonal matrices. This assumption normalizes gradients by scaling both rows and columns, leading to the RACS algorithm. It results in a good structural approximation of a nearly block diagonal Hessian matrix in transformer-based networks with mean-square error (MSE) loss and cross-entropy (CE) loss; see (Dong et al., 2025). Then minimizing the objective $\min_{\tilde{F}} \|\tilde{F} - F\|_F^2$ will give the following equation:

$$\tilde{F} = S \otimes Q, \quad s = \frac{\text{Diag}\left(\mathbb{E}\left[G^T Q G\right]\right)}{\|Q\|_F^2}, \quad q = \frac{\text{Diag}\left(\mathbb{E}\left[G S G^T\right]\right)}{\|S\|_F^2}, \tag{14}$$

where $s = \text{Diag}(S), q = \text{Diag}(Q)$. RACS (Gong et al., 2025) uses a one-sample estimate of $\mathbb{E}[\cdot]$ and performs a 5-step power iteration to solve (14), starting from $q = 1$. And they only accumulate the first moments of $s$ and $q$, reducing the memory consumption to $m + n$, that is:

$$m_{s_t} = \beta_1 s_{t-1} + (1 - \beta_1) s_t, \quad m_{q_t} = \beta_1 q_{t-1} + (1 - \beta_1) q_t.$$

Incorporating these moments with the same descent vector as in AdamS (Algorithm 2), we propose LRACS, short for "Lifted RACS". The iteration scheme of LRACS is provided in Appendix (A.10). We provide a runtime analysis about the two algorithms in section 5.

## 5 EXPERIMENTAL RESULTS

We evaluated the proposed LoRA optimizers against other optimizers in transformer-based models with different loss functions. These include LoRA fine-tuning on the Mix-of-show (Gu et al., 2023) image generation model, evaluated using the clip-vit-base-patch16 model (Radford et al., 2021), and fine-tuning a GPT-2 medium LLM (Radford et al., 2019) on the E2E dataset (Novikova et al., 2017) for the natural language generation challenge. The results are shown in Table 1 and Table 2.

|  | RANK | CLIP ↑ | FID ↓ |  | RANK | CLIP ↑ | FID ↓ |
|---|---|---|---|---|---|---|---|
| SGD | 4 | 23.90 | 77.66 | Adam | 4 | 25.17 | 72.75 |
|  | 8 | 24.52 | 74.00 |  | 8 | 26.83 | 67.09 |
|  | 16 | 24.76 | 70.78 |  | 16 | 29.86 | 57.35 |
| Scaled GD | 4 | 25.74 | 70.99 | Quotient GD | 4 | 24.34 | 75.48 |
|  | 8 | 25.98 | 70.01 |  | 8 | 24.46 | 74.91 |
|  | 16 | 29.07 | 59.87 |  | 16 | 28.04 | 63.20 |
| AdamW | 4 | 27.86 | 64.38 | LRACS (ours) | 4 | 31.43 | 52.67 |
|  | 8 | 28.05 | 62.58 |  | 8 | 31.46 | 52.12 |
|  | 16 | 30.78 | 62.57 |  | 16 | 32.09 | 49.52 |
| LoRA-Rite | 4 | 30.96 | 59.04 | AdamS (ours) | 4 | 32.20 | 54.39 |
|  | 8 | 30.99 | 59.02 |  | 8 | 32.38 | 51.87 |
|  | 16 | 31.90 | 55.68 |  | 16 | **32.64** | **46.32** |

Table 1: Experimental results of Mix-of-show generation model on CLIP and FID score.

We compare the following optimizers:

- SGD (Amari, 1993): One of the most widely used optimizer in machine learning. It is conventional and does not support EFL.
- Adam (Kingma & Ba, 2014) and AdamW (Loshchilov & Hutter, 2017): The most widely used optimizers in vanilla LoRA. The vanilla Adam does not support weight-decay, while AdamW needs more finetuning about the weight-decay parameter. They are conventional and do not support EFL.
- Scaled GD (Tong et al., 2021) and Quotient GD (Mishra et al., 2014): Both are Riemannian optimization methods which support EFL and no extra preconditioning is applied.

| Method | BLEU | NIST | MET | ROUGE-L | CIDEr |
|---|---|---|---|---|---|
| SGD$_{r=4}$ | 66.6 | 8.54 | 44.2 | 68.2 | 2.32 |
| Adam$_{r=4}$ | 68.0 | 8.61 | 44.7 | 69.1 | 2.38 |
| Scaled GD$_{r=4}$ | 68.5 | 8.72 | 45.5 | 69.4 | 2.40 |
| Quotient GD$_{r=4}$ | 66.9 | 8.54 | 44.3 | 68.4 | 2.35 |
| AdamW$_{r=4}$ | 68.6 | 8.69 | 46.5 | 71.3 | 2.51 |
| LoRA-Rite$_{r=4}$ | 69.3 | 8.75 | 46.5 | 71.7 | 2.53 |
| AdamS (ours)$_{r=4}$ | 69.4 | 8.75 | 46.5 | 71.7 | 2.53 |
| LRACS (ours)$_{r=4}$ | **70.4** | **8.85** | **46.7** | **71.9** | **2.54** |

Table 2: Experimental results of GPT-2 medium LLM on E2E NLG challenge.

- LoRA-Rite (Yen et al., 2025): The state-of-the-art optimizer supports EFL.

- LRACS (Algorithm 3.): Our proposed algorithm which is constructed by lifting RACS (Gong et al., 2025) to horizontal space. Following the general scheme, it supports EFL.

- AdamS (Algorithm 2): Our proposed algorithm which is constructed by lifting Adam to horizontal space. Following the general scheme, it supports EFL.

**Optimizer setup.** We perform a grid search to select the best learning rate for each algorithm, with the optimal hyperparameters listed in Appendix (A.2). For AdamS, we implement the low-memory variant (Algorithm 4) and enable the Riemannian inner product option. For LRACS, following (Gong et al., 2025), we train the last LoRA layer with Adam (Kingma & Ba, 2014) if it is injected into the last layer of the network. All experiments are run on an A100 GPU. Additional details on robust training are provided in Appendix (A.16).

**Results of Mix-of-show model.** Mix-of-show is a transformer-based model with Mean-Square Error (MSE) loss. Thus, it admits block diagonal Hessian structure (Dong et al., 2025). We used the same image dataset as in the original Mix-of-Show (Gu et al., 2023) paper—the Harry Potter image set—and trained for 3,500 steps. We then saved the generated images (see the Appendix) and computed their CLIP and FID (Parmar et al., 2022) scores. A higher CLIP score indicates a stronger relevance between the generated images and the text descriptions, while a lower FID signifies that the generated images possess more distinctive features of a specific category. As demonstrated in Table 1, methods that achieve efficient feature learning (EFL) generally outperform the corresponding conventional approaches that do not. At $r = 4$, compared to the non-EFL baseline, our proposed method AdamS improves the CLIP score from 25.17 (Adam) to 32.20 and reduces the FID score from 72.75 (Adam) to 54.39. At $r = 16$, compared to the EFL baseline, AdamS outperforms Scaled GD, raising CLIP from 29.07 to 32.64 and lowering FID from 59.87 to 46.32. These results demonstrate substantial gains over the baselines. Moreover, our method demonstrates superior performance over state-of-the-art approaches, i.e. LoRA-Rite, on key benchmark metrics. AdamS consistently achieves the highest CLIP scores (above 32) for $r = 4, 8, 16$, indicating both superior quality and robustness.

**Results of GPT-2.** GPT-2 is a transformer-based model with Cross Entropy (CE) loss. It also admits block diagonal Hessian structure. For experiment in GPT-2 medium, we choose $r = 4$ for LoRA based on the previous E2E NLG challenge study (Zhang & Pilanci, 2024).The E2E dataset is a dataset for training end-to-end, data-driven natural language generation systems in the restaurant domain. The dataset is split into training, validation and testing sets (in a 76.5-8.5-15 ratio). Success on this task is typically measured by achieving a high BLEU (Papineni et al., 2002), NIST, METEOR (Banerjee & Lavie, 2005), Rouge-L (Lin, 2004), CIDEr. In the GPT-2 medium experiments, the total number of trainable parameters is 0.39M. We train on the dataset for 5 epochs, totaling 22,600 steps. The training hyperparameters are documented in the Appendix (A.2). As demonstrated in Table 2, it can be observed that, the EFL version of Adam, AdamS generally outperform Adam on all evaluation metrics. Within the EFL methods, LRACS achieves the best performance demonstrating significant improvements over baseline methods as well as comprehensive superiority over other state-of-the-art approaches. We believe that this characteristic will greatly inspire future research. To enrich the diversity and scalability results, we also conducted experiments on WebNLG dataset and the GPT-2 small model, which is recorded in Appendix (A.14).

**Ablation Study.** We tested $r = 4, 8, 16$ on the image generation model to ensure the robustness to rank of our proposed methods. It turned out that $r = 16$ is the optimal rank value determined experimentally in Mix-of-show model. Besides, in Section 3.3 and Appendix (A.6), we also conducted an ablation study replacing the solution of the Sylvester equation with a weight decay matrix. The results show that more accurate approximations to the Sylvester equation lead to better performance, indicating that solving the Sylvester equation is indispensable. In the Appendix (A.11), we performed an ablation study on regularizing the Sylvester matrix K, and the results show that regularizing the Sylvester matrix does not improve performance. Finally, we also conduct an ablation study about Riemannian and Euclidean momentum accumulation on GPT-2 small model in Appendix (A.13).

**Sensitivity Analysis.** We conducted a detailed sensitivity analysis of the hyperparameters (learning rate & batch size) for AdamS and LRACS at $r = 4$ on GPT-2 medium model. We first fix the learning rate of AdamS to $2e^{-4}$ and LRACS to $2e^{-3}$ then vary it from 50% to 250%. A similar result about varying batch size in Mix-of-show model is presented in the Appendix (A.15). The results demonstrate that AdamS and LRACS are robust to these changes.

| Method | learning rate | WebNLG | | | | | | | | |
| --- | --- | --- | --- | --- | --- | --- | --- | --- | --- | --- |
| | | BLEU ↑ | | | chrf++ ↑ | | | TER ↓ | | |
| | | U | S | A | U | S | A | U | S | A |
| AdamS | $1e^{-4}$ | 43.40 | 61.72 | 53.62 | . 62 | **. 74** | . 68 | . 49 | .34 | **.41** |
| | $2e^{-4}$ | 42.95 | **62.73** | 53.92 | . 62 | **. 74** | **. 69** | . 49 | .34 | **.41** |
| | $5e^{-4}$ | 43.28 | 62.38 | 53.42 | . 62 | **. 74** | . 68 | . 49 | .34 | **.41** |
| LRACS | $1e^{-3}$ | 43.51 | 61.99 | 53.75 | **. 63** | **. 74** | **. 69** | . 49 | .34 | **.41** |
| | $2e^{-3}$ | **44.02** | 62.42 | **53.94** | . 62 | **. 74** | **. 69** | **. 48** | **.33** | **.41** |
| | $5e^{-3}$ | 43.29 | 61.46 | 53.39 | . 62 | **. 74** | **. 69** | . 49 | .34 | **.41** |

Table 3: Results of optimizers on GPT-2 medium model in sensitivity analysis. The results show that our proposed methods are robust to hyperparameter changes

**Time and Space Complexity** In each iteration, the dominant computational costs for **AdamS** (Algorithm 2) are as follows: (1) matrix inverses for $r$-by-$r$ matrices which takes $\mathcal{O}(r^3)$ time, (2) solving Sylvester equation (9) for $r$-by-$r$ matrices which takes $\mathcal{O}(r^3)$ time, (3) projections of $n$-by-$m$ matrices which take $\mathcal{O}(nr^2 + mr^2 + mnr)$ time and (4) matmuls with time complexity $\mathcal{O}(mr^2 + nr^2)$. Additionally, the backpropagation step for the loss function is computationally intensive, requiring at least $\Omega(mnr)$ flops to compute the chain rule: $\nabla M = \nabla W N, \nabla N = \nabla W^{\mathrm{T}} M$. Therefore, the overall per-step complexity for **AdamS** is $\Omega(mnr) + \mathcal{O}(r^3 + mr^2 + nr^2)$. Similarly, the per-step complexity for **LRACS** (Algorithm 3) is $\Omega(mnr) + \mathcal{O}(r^3 + mr^2 + nr^2)$. In comparison, standard AdamW has per-step complexity $\Omega(mnr) + \mathcal{O}(mr^2 + nr^2)$. As for the memory cost due to moments storage, **AdamS** takes $\mathcal{O}(mn)$ while **LRACS** takes $\mathcal{O}(m + n + 1)$. Meanwhile, AdamW needs $\mathcal{O}(mr + nr)$ for moments storage. For memory consumption and training time per step in real experiments, see the table (7) in Appendix (A.12).

## 6 Conclusions

Current LoRA optimization methods lack emphasis on efficient feature learning (EFL). The few existing EFL methods are hard to generalize. This paper proposes LoRA-S, which is a framework applying horizontal lift theory to bridge pre-training optimizers (Adam and RACS) and lifting LoRA methods (AdamS and LRACS) that admit efficient feature learning. The new algorithms are weight-decay-free, indicating less hyperparameter tuning. In empirical tasks, our algorithms consistently achieve better results than existing optimizers in diverse datasets and transformer-based models. Because our contribution is a mathematical, theory-driven method for optimizer design, the conclusions of this paper remain valid even in complex engineering scenarios such as distributed computing and larger-scale LLMs with reasonable additional computational overhead. We hope that the framework and theoretical analysis presented in this work will help in designing and iterating EFL algorithms more efficiently and rationally.

ACKNOWLEDGMENTS AND FUNDING DISCLOSURE

We thank the anonymous reviewers for their valuable suggestions on theory and experiments. This work is partially supported by Hong Kong Research Grant Council GRFs 16307023 and 16306124.

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

# A  APPENDIX

## A.1  LARGE LANGUAGE MODELS (LLMs) IN PAPER WRITING

LLM is used in this paper to improve grammar and wording.

## A.2  HYPERPARAMETERS

| Method | AdamW | | | LoRA-Rite | | | AdamS | | | LRACS | | |
|---|---|---|---|---|---|---|---|---|---|---|---|---|
| Rank | 4 | 8 | 16 | 4 | 8 | 16 | 4 | 8 | 16 | 4 | 8 | 16 |
| **Training** | | | | | | | | | | | | |
| Weight decay | | 0.01 | | | 0.0 | | | / | | | / | |
| Batch Size | | | | | | 2 | | | | | | |
| LR Scheduler | | | | | | Linear | | | | | | |
| LR (tuned, $\times 10^{-3}$) | 1e-2 | 1e-2 | 2e-2 | 1 | 1 | 1 | 0.2 | 0.2 | 0.3 | 1 | 3 | 5 |
| $1^{st}$ Momentum decay $\beta_1$ | | 0.9 | | | 0.9 | | | 0.9 | | | 0.9 | |
| $2^{nd}$ Momentum decay $\beta_2$ | | 0.999 | | | 0.999 | | | 0.999 | | | / | |
| **Inference** | | | | | | | | | | | | |
| Inference Steps | | | | | | 50 | | | | | | |
| Guidance Scale | | | | | | 7.5 | | | | | | |
| Fusion parameter $\alpha$ | | | | | | 0.7 | | | | | | |

Table 4: Hyperparameters for Mix-of-show model fine-tuning.

| Method | AdamW | LoRA-Rite | AdamS | LRACS |
|---|---|---|---|---|
| Rank | 4 | 4 | 4 | 4 |
| **Training** | | | | |
| Weight decay | 0.01 | 0.0 | / | / |
| Train batch size | | 8 | | |
| LR Scheduler | | Linear | | |
| Warmup step | | 500 | | |
| Label smooth | | 0.1 | | |
| LoRA $\alpha$ | | 32 | | |
| LoRA dropout | | 0.1 | | |
| LR (tuned) | 2e-4 | 1e-3 | 5e-5 | 1e-3 |
| $1^{st}$ Momentum decay $\beta_1$ | 0.9 | 0.9 | 0.9 | 0.9 |
| $2^{nd}$ Momentum decay $\beta_2$ | 0.999 | 0.999 | 0.999 | / |
| **Inference** | | | | |
| Beam Size | | 10 | | |
| Length Penalty | | 0.8 | | |
| No Repeat Ngram Size | | 4 | | |

Table 5: Hyperparameters for GPT-2 model fine-tuning.

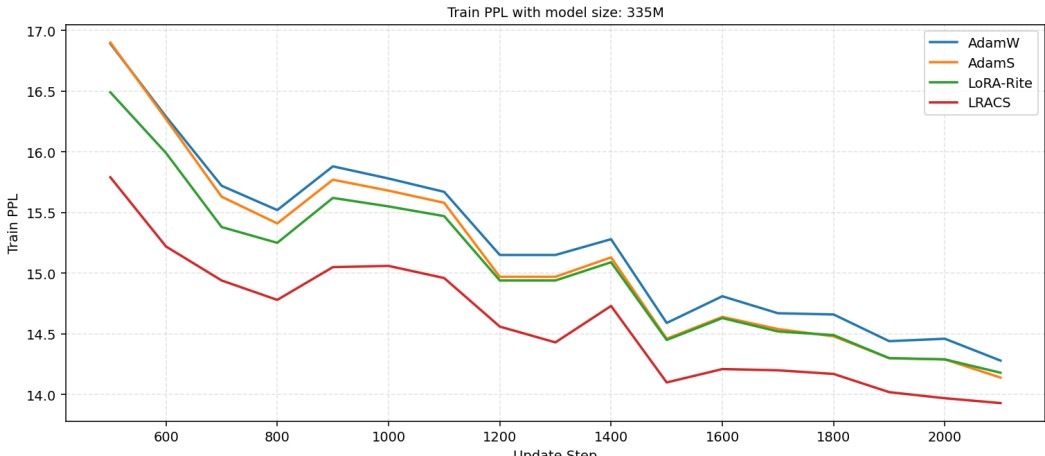

Figure 2: 335M GPT-2 medium model LoRA training PPL curve begins from warmup steps.

### A.3 TRAINING LOSS CURVE VISUALIZATION

### A.4 INDUCED METRIC FOR A TRANSFORMATION INVARIANCE SCHEME (PROPOSITION 1)

Recall the LoRA-Rite update scheme (Yen et al., 2025):

$$\delta M_1 = \bar{\nabla} M_1 \left(\bar{\nabla} M_1^\top \bar{\nabla} M_1\right)^{-\frac{1}{2}} \left(R_{N_1}^\top\right)^\dagger$$

$$\delta M_2 = \bar{\nabla} M_2 \left(\bar{\nabla} M_2^\top \bar{\nabla} M_2\right)^{-\frac{1}{2}} \left(R_{N_2}^\top\right)^\dagger$$

Then the first part of our inner product $\bar{g}_{(M,N)}$ becomes

$$\text{trace}(N_1^\text{T} N_1 \delta M_1^\text{T} \delta M_1) = \text{trace}(R_{N_1}^\text{T} R_{N_1} R_{N_1}^\dagger \left(\bar{\nabla} M_1^\top \bar{\nabla} M_1\right)^{-\frac{1}{2}} \bar{\nabla} M_1^\top \bar{\nabla} M_1 \left(\bar{\nabla} M_1^\top \bar{\nabla} M_1\right)^{-\frac{1}{2}} (R_{N_1}^\text{T})^\dagger)$$

$$= \text{trace}(\boldsymbol{I}_r) = \text{trace}(N_2^\text{T} N_2 \delta M_2^\text{T} \delta M_2) \tag{15}$$

Similarly, we have symmetric results for $\delta N_1$ and $\delta N_2$ (the second part):

$$\text{trace}(M_1^\text{T} M_1 \delta N_1^\text{T} \delta N_1) = \text{trace}(\boldsymbol{I}_r) = \text{trace}(M_2^\text{T} M_2 \delta N_2^\text{T} \delta N_2) \tag{16}$$

Equations (15) and (16) lead to the induced metric:

$$\bar{g}_{(M_1,N_1)} = ((\delta M_1, \delta N_1), (\delta M_1, \delta N_1)) = \text{trace}(N_1^\text{T} N_1 \delta M_1^\text{T} \delta M_1 + M_1^\text{T} M_1 \delta N_1^\text{T} \delta N_1)$$

$$= \text{trace}(N_2^\text{T} N_2 \delta M_2^\text{T} \delta M_2 + M_2^\text{T} M_2 \delta N_2^\text{T} \delta N_2) = \bar{g}_{(M_2,N_2)}$$

Since $(\delta M_i, \delta N_i), i = 1, 2$ are preconditioned with AdaGrad (Ward et al., 2020), LoRA-Rite can be viewed as lifting AdaGrad under this specific metric.

### A.5 PROOF OF TRANSFORMATION INVARIANCE (THEOREM 1)

(Yen et al., 2025) proposed the definition of transformation invariance and all optimizers following the general iteration scheme (Algorithm 1) are transformation invariance. Let $K_0$ be the Sylvester matrix in the equation

$$M^\text{T} \dot{X}_{MN^\text{T}} N = M^\text{T} M N^\text{T} N K_0 + K_0 M^\text{T} M N^\text{T} N$$

And $K_1$ be the solution of the equation

$$(MR)^\text{T} \dot{X}_{MN^\text{T}} (NR^{-\text{T}}) = (MR)^\text{T} (MR)(NR^{-\text{T}})^\text{T} N R^{-\text{T}} K_1 + K_1 (MR)^\text{T} MR (NR^{-\text{T}})^\text{T} N R^{-\text{T}} \tag{17}$$

in which $R \in \text{GL}(r)$. Note that the equation (17) can be reduced to

$$M^\text{T} \dot{X}_{MN^\text{T}} N = M^\text{T} M N^\text{T} N R^{-\text{T}} K_1 R^\text{T} + R^{-\text{T}} K_1 R^\text{T} M^\text{T} M N^\text{T} N$$

Thus we gain the relation between $K_0$ and $K_1$

$$K_0 = R^{-\mathrm{T}} K_1 R^{\mathrm{T}} \tag{18}$$

holds for all $t = 1, 2, ..., T$. Now we look at the iteration scheme for a particular time $t$, and LoRA pair $(M_t, N_t)$, (Algorithm 1) gives the following:

$$M_{t+1} = M_t + \eta \dot{X}_{\uparrow \mathrm{M}(M_t, N_t)} \text{ and } N_{t+1} = N_t + \eta \dot{X}_{\uparrow \mathrm{N}(M_t, N_t)}$$

And the iteration scheme for another LoRA pair $(M_t R, N_t R^{-\mathrm{T}})$ is

$$M'_{t+1} = M_t R + \eta \dot{X}_{\uparrow \mathrm{M}(M_t R, N_t R^{-\mathrm{T}})} \text{ and } N'_{t+1} = N_t R^{-\mathrm{T}} + \eta \dot{X}_{\uparrow \mathrm{N}(M_t R, N_t R^{-\mathrm{T}})}$$

To prove the transformation invariance property, just observe that

$$\dot{X}_{\uparrow \mathrm{M}(M_t R, N_t R^{-\mathrm{T}})} = (\dot{X}_{MN^{\mathrm{T}}} N_t R^{-\mathrm{T}} - M_t N_t^{\mathrm{T}} N_t R^{-\mathrm{T}} K_t^{(1)}) R^{\mathrm{T}} (N_t^{\mathrm{T}} N_t)^{-1} R$$

$$= (\dot{X}_{MN^{\mathrm{T}}} N_t - M_t N_t^{\mathrm{T}} N_t R^{-\mathrm{T}} K_t^{(1)} R^{\mathrm{T}}) (N_t^{\mathrm{T}} N_t)^{-1} R$$

From equation (18), we can substitute $K_t^{(1)}$ with $K_t^{(0)}$, thus

$$\dot{X}_{\uparrow \mathrm{M}(M_t R, N_t R^{-\mathrm{T}})} = \dot{X}_{\uparrow \mathrm{M}(M_t, N_t)} R$$

We can derive similar result for $\dot{X}_{\uparrow \mathrm{N}(M_t R, N_t R^{-\mathrm{T}})}$

$$\dot{X}_{\uparrow \mathrm{N}(M_t R, N_t R^{-\mathrm{T}})} = \dot{X}_{\uparrow \mathrm{N}(M_t, N_t)} R^{-\mathrm{T}}$$

These relations of the lift vectors lead to the following

$$\dot{X}_{\uparrow \mathrm{M}(M_t, N_t)} N_t^{\mathrm{T}} = \dot{X}_{\uparrow \mathrm{M}(M_t R, N_t R^{-\mathrm{T}})} R^{-1} N_t^{\mathrm{T}}$$
$$M_t \dot{X}_{\uparrow \mathrm{N}(M_t, N_t)}^{\mathrm{T}} = M_t R \dot{X}_{\uparrow \mathrm{N}(M_t R, N_t R^{-\mathrm{T}})}^{\mathrm{T}}$$
$$\dot{X}_{\uparrow \mathrm{M}(M_t, N_t)} \dot{X}_{\uparrow \mathrm{N}(M_t, N_t)}^{\mathrm{T}} = \dot{X}_{\uparrow \mathrm{M}(M_t R, N_t R^{-\mathrm{T}})} \dot{X}_{\uparrow \mathrm{N}(M_t R, N_t R^{-\mathrm{T}})}^{\mathrm{T}}$$

which leads to the transformation invariance condition

$$M_{t+1} N_{t+1}^{\mathrm{T}} = M'_{t+1} (N'_{t+1})^{\mathrm{T}} \tag{19}$$

## A.6 RELATIVE ERROR OF SGD OPTIMIZER

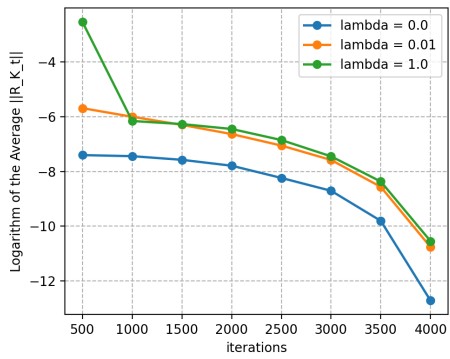
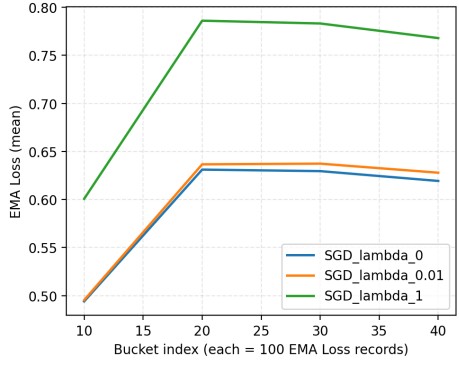

(a) Relative error of SGD for $\lambda = 0, 0.01, 1$      (b) EMA Loss of SGD for $\lambda = 0, 0.01, 1$

Figure 3: Weight decay $\lambda$ that yields lower relative error tends to achieve lower EMA loss.

## A.7 A PRACTICAL INTUITION FOR WEIGHT-DECAY MATRIX VS SYLVESTER MATRIX

We have already made an experiment showing that weight-decay matrix that is closer to K achieves a lower loss in Fig (3) and Fig (3.3). Now we provide two complementary perspectives for intuition.

**Perspective: Update Norm**

In fact, another interpretation of efficient feature learning studies the update norm divided by the weight norm, i.e. $\|\delta M\|/\|M\|$ and $\|\delta N\|/\|N\|$, see LoRA-Rite's (Yen et al., 2025) (A.8). Recall the general iteration scheme $\delta M = GT_M + DT_M, \delta N = GT_N + DT_N$, the gradient terms are $GT_M = \dot{X}N\left(N^T N\right)^{-1}, GT_N = \dot{X}^T M\left(M^T M\right)^{-1}$. When applying weight-decay matrix, the decay terms are $DT_M(\epsilon) = -\epsilon M, DT_N(\epsilon) = -\epsilon N$. When applying sylvester matrix, they are

$$
DT_M = -M\left(N^T N\right) K\left(N^T N\right)^{-1}, DT_N = -N\left(M^T M\right) K^T\left(M^T M\right)^{-1},
$$

One can find that the magnitudes of both $\|GT_M\|/\|M\|$ and $\|DT_M\|/\|M\|$ are $O(\|K\|)$ while the magnitude of $\|DT_M(\epsilon)\|/\|M\|$ is $O(\epsilon)$ which results in a magnitude gap as $\epsilon$ is a hyperparameter. For example, in weight-decay matrix case, when $\epsilon >> \|GT_M\|/\|M\|$, $\|\delta M\|/\|M\| = O(\epsilon)$. The preconditioning information in $GT_M$ cannot be properly reflected in the updates, which leads to less efficient learning. Similar results also appear in $\|\delta N\|/\|N\|$.

**Perspective: Preconditioning**

Our preconditioning vector $\dot{X}$ mitigates the condition number of the Hessian $\nabla^2 f_L(W)$ efficiently during large-model training, which is also an important reason for the success. When applying the Sylvester matrix, the overall update is $\delta M N^T + M \delta N^T = \dot{X}$. However, applying weight-decay matrix will undermine the preconditioning effect. As the overall update becomes:

$$
\delta M N^T + M \delta N^T = \dot{X} N N^+ + M M^+ \dot{X} - 2\epsilon M N^T
$$

In this case, the matrices $MN^T, NN^+$ and $MM^+$ are likely to be ill-conditioned, making the preconditioning less effective. Thus, the Sylvester-based update tends to be more efficient which indicates the weight-decay parameter in our methods can be removed.

A.8    COMPUTATION OF PROJECTION

Mathematically, we need to project $\tilde{X}_{MN^T}$ back to $\mathrm{T}_{M_t N_t^T}$ using the following:

$$
\mathrm{P}^{(e)}_{\mathrm{T}_{M_t N_t^T}}(\tilde{X}_{MN^T}) = UU^{\mathrm{T}}\tilde{X}_{MN^T} + \tilde{X}_{MN^T}VV^T - UU^{\mathrm{T}}\tilde{X}_{MN^T}VV^T.
$$

However, the projection can be omitted since the update rule for $(M_t, N_t)$ (lines 4 and 5 in Algorithm 1) will eliminate the components of $\tilde{X}_{MN^T}$ orthogonal to $\mathrm{T}_{M_t N_t^T}$. Consider $\tilde{X}_{MN^T}N$ and $\tilde{X}^{\mathrm{T}}_{MN^T}M$ in "Gradient Terms", we have:

$$
\tilde{X}_{MN^T} = \mathrm{P}^{(e)}_{\mathrm{T}_{M_t N_t^T}}(\tilde{X}_{MN^T}) + (\boldsymbol{I} - UU^{\mathrm{T}})\tilde{X}_{MN^T}(\boldsymbol{I} - VV^{\mathrm{T}}),
$$

$$
\tilde{X}_{MN^T}N = \mathrm{P}^{(e)}_{\mathrm{T}_{M_t N_t^T}}(\tilde{X}_{MN^T})N + (\boldsymbol{I} - UU^{\mathrm{T}})\tilde{X}_{MN^T}(\boldsymbol{I} - VV^{\mathrm{T}})VR_N = \mathrm{P}^{(e)}_{\mathrm{T}_{M_t N_t^T}}(\tilde{X}_{MN^T})N,
$$

$$
\tag{20}
$$

$$
\tilde{X}^{\mathrm{T}}_{MN^T}M = (\mathrm{P}^{(e)}_{\mathrm{T}_{M_t N_t^T}}(\tilde{X}_{MN^T}))^{\mathrm{T}}M + (\boldsymbol{I} - VV^{\mathrm{T}})(\tilde{X}_{MN^T})^{\mathrm{T}}(\boldsymbol{I} - UU^{\mathrm{T}})UR_M
$$

$$
= (\mathrm{P}^{(e)}_{\mathrm{T}_{M_t N_t^T}}(\tilde{X}_{MN^T}))^{\mathrm{T}}M,
\tag{21}
$$

where $U$ and $V$ are the orthonormal bases from the QR decompositions of $M$ and $N$, i.e., $M = UR_M, N = VR_N$. The tricks in (20)-(21) significantly reduce the computational cost of projection.

A.9 ITERATION SCHEME OF ADAMS

---

**Algorithm 2 AdamS**

---

1: **Input:** learning rate $\lambda$, first momentum decay $\beta_1$, second momentum decay $\beta_2$, optimization steps $T$. **Init** with $m_0 = 0; v_0 = 0$
2: **for** $t = 1 \ldots T$ **do**
3:     Obtain the descent vector $\dot{X}_{M_t N_t^\mathrm{T}}$ from (12)
4:     Calculate first momentum $m_t \leftarrow \beta_1 m_{t-1} + (1 - \beta_1)\dot{X}_{M_t N_t^\mathrm{T}}$
5:     Calculate second momentum $v_t \leftarrow \beta_2 v_{t-1} + (1 - \beta_2)\dot{X}^2_{M_t N_t^\mathrm{T}}$
6:     Calculate preconditioned vector $\tilde{X}_{M_t N_t^\mathrm{T}} \leftarrow m_t / \sqrt{v_t}$
7:     Solve $K_t$ from the Sylvester equation (9) to lift $\tilde{X}_{M_t N_t^\mathrm{T}}$
8:     Compute the lift vector $\dot{X}_{\uparrow\mathrm{M}(M_t, N_t)} \leftarrow \left( \tilde{X}_{M_t N_t^\mathrm{T}} N_t - M_t N_t^\mathrm{T} N_t K_t \right) \left( N_t^\mathrm{T} N_t \right)^{-1}$
9:     Compute the lift vector $\dot{X}_{\uparrow\mathrm{N}(M_t, N_t)} \leftarrow \left( \tilde{X}^\mathrm{T}_{M_t N_t^\mathrm{T}} M_t - N_t M_t^\mathrm{T} M_t K_t^\mathrm{T} \right) \left( M_t^\mathrm{T} M_t \right)^{-1}$
10:     Update model parameters for $(M_{t+1}, N_{t+1}) \leftarrow (M_t + \dot{X}_{\uparrow\mathrm{M}(M_t, N_t)}, N_t + \dot{X}_{\uparrow\mathrm{N}(M_t, N_t)})$
11: **end for**

---

A.10 ITERATION SCHEME OF LRACS

---

**Algorithm 3 LRACS**

---

1: **Input:** learning rate $\lambda$, first momentum decay $\beta_1$, limiter threshold $\gamma$, optimization steps $T$.
2: **Init:** $m_{s_0} = 0; m_{q_0} = 0; \phi_0 = 0$
3: **for** $t = 1, \ldots, T$ **do**
4:     Obtain the projected gradient $G$ from (12)
5:     Obtain $s_t$ and $q_t$ from (14)
6:     Calculate momentums: $m_{s_t} \leftarrow \beta_1 m_{s_{t-1}} + (1 - \beta_1)s_t, m_{q_t} \leftarrow \beta_1 m_{q_{t-1}} + (1 - \beta_1)q_t$
7:     Calculate preconditioned vector: $\tilde{X}_{M_t N_t^\mathrm{T}} \leftarrow \mathrm{Diag}_v(m_{q_t})^{-1/2} G \mathrm{Diag}_v(m_{s_t})^{-1/2}$
8:     Solve $K_t$ from the Sylvester equation (9) to lift $\tilde{X}_{M_t N_t^\mathrm{T}}$
9:     Compute the lift vector $\dot{X}_{\uparrow\mathrm{M}(M_t, N_t)} \leftarrow \left( \tilde{X}_{M_t N_t^\mathrm{T}} N_t - M_t N_t^\mathrm{T} N_t K_t \right) \left( N_t^\mathrm{T} N_t \right)^{-1}$
10:     Compute the lift vector $\dot{X}_{\uparrow\mathrm{N}(M_t, N_t)} \leftarrow \left( \tilde{X}^\mathrm{T}_{M_t N_t^\mathrm{T}} M_t - N_t M_t^\mathrm{T} M_t K_t^\mathrm{T} \right) \left( M_t^\mathrm{T} M_t \right)^{-1}$
11:     Incorporate the norm-growth limiter:
12:     $\eta = \gamma / \max \left\{ \left\| \tilde{X}_{M_t N_t^\mathrm{T}} \right\|_g / \phi_{t-1}, \gamma \right\}$ if $t > 1$ else $1, \phi_t = \eta \left\| \tilde{X}_{M_t N_t^\mathrm{T}} \right\|_g$
13:     Update model parameters for $(M_{t+1}, N_{t+1}) \leftarrow (M_t - \lambda\eta\dot{X}_{\uparrow\mathrm{M}(M_t, N_t)}, N_t - \lambda\eta\dot{X}_{\uparrow\mathrm{N}(M_t, N_t)})$
14: **end for**

---

A.11 EFFECT OF REGULARIZATION

As explained in Section 3.3, we can apply weight decay together with the Sylvester matrix. However, this regularization will introduce computational error for Sylvester equation (9) then further influence the effect of efficient feature learning. Regularize $K_t$ with the weight decay parameter $\lambda$ and the learning rate $\eta$:

$$K_t \leftarrow K_t + \lambda\eta \boldsymbol{I}_r \tag{22}$$

Then we conduct experiments on Mixofshow model with AdamS (Algorithm.2) optimizer. We test the influence of $\lambda$ on an empirical task, the CLIP score, since $L_2$ regularization mainly addresses the overfitting problem. The results show that the method with the least regularization achieves the highest CLIP score. Thus, empirically, we have the confidence to replace the weight decay matrix with the Sylvester decay matrix $K_t$ directly.

|  | $\lambda = 0.0$ | $\lambda = 0.01$ | $\lambda = 0.1$ | $\lambda = 1.0$ |
|---|---|---|---|---|
| CLIP | **32.64** | 32.28 | 31.97 | 32.29 |

Table 6: Regularization of Sylvester matrix K will not improve performance.

|  | AdamW | LoRA-Rite | LRACS | AdamS |
|---|---|---|---|---|
| Memory (MiB) | 10490 | 10530 | 10522 | 10546 |
| Time (ms/batch) | 569 | 767 | 734 | 653 |

Table 7: Memory consumption and training time per step in Mix-of-show model.

## A.12 TIME AND SPACE COMPLEXITY

## A.13 ABLATION STUDY IN RIEMANNIAN AND EUCLIDEAN MOMENTUM

## A.14 DIVERSITY AND SCALABILITY RESULTS

Besides GPT-2 medium model with the E2E dataset, to enrich the scalability of our experimental results, we conducted experiments on the GPT-2 small model with E2E dataset. To enrich the diversity of our experiments, we conduct experiments on the GPT-2 medium model, WebNLG (Gardent et al., 2017) dataset. WebNLG is a popular dataset for data-to-text evaluation which includes 22K examples from 14 distinct categories. The validation set is spitted into U (unseen), S (seen) and A (all). The results show that our proposed method outperforms current methods across all metrics and we have recorded the results in Table (8).

| Method | E2E | | | | |
|---|---|---|---|---|---|
|  | BLEU | NIST | MET | ROUGE-L | CIDEr |
| $Adam_{r=4}$ | 68.8 | 8.71 | 46.0 | 70.3 | 2.45 |
| $LoRA\text{-}Rite_{r=4}$ | 69.1 | 8.75 | 46.1 | 70.5 | 2.47 |
| $AdamS_{r=4}$ | 69.4 | 8.75 | **46.5** | 71.3 | **2.51** |
| $LRACS_{r=4}$ | **69.7** | **8.79** | 46.4 | **71.5** | 2.50 |

Table 8: Results of optimizers on GPT-2 small model in E2E NLG challenge. The results show that our proposed method outperforms current methods across all metrics.

## A.15 SENSITIVITY ANALYSIS

## A.16 ROBUST TRAINING

In this section, we enumerate the challenges encountered during training and the corresponding solutions. As a variant of Riemannian optimization algorithm, LoRA-S faces similar training failure cases. For example, the computation of $(M^{\mathrm{T}}M)^{-1}$ is inaccurate under the case of zero initialization. The same issue also arises in LoRA-Rite(Yen et al., 2025), ScaledGD, and ScaledAdam (Zhang & Pilanci, 2024). Here, we list approaches from other works as well as our proposed methods to ensure robust training of LoRA-S. Among them, we implement the zero initialization (zero-init) in our experiments.

**Zero Initialization.** Zero initialization is used most common in LoRA training. In this start, the first LoRA factor is initialized with normal distribution with mean 0 and variance $\alpha^2$, that is:

$$[\boldsymbol{M}_0]_{ij} \sim \mathcal{N}\left(0, \alpha^2\right) \quad [\boldsymbol{N}_0]_{ij} = 0, \quad \alpha > 0. \text{ (zero-init)}$$

In this circumstance, the computation of $(N_0^{\mathrm{T}}N_0)^{-1}$ explodes and the loss becomes $nan$, which makes the whole training inefficient. In response to this situation, other methods take the following

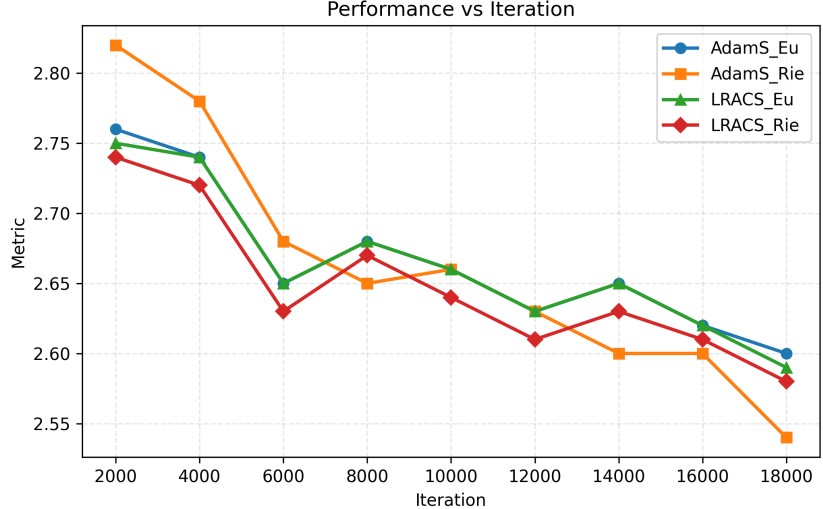

Figure 4: Results of train loss of our optimizers on GPT2 small model in E2E NLG challenge with different momentums. The results show that Riemannian momentum accumlation provides a slight improvement for AdamS, but not for LRACS.

| Method | BLEU ↑ | | | chrf++ ↑ | | | TER ↓ | | |
| | WebNLG | | | | | | | | |
| --- | --- | --- | --- | --- | --- | --- | --- | --- | --- |
| | U | S | A | U | S | A | U | S | A |
| $Adam_{r=4}$ | 41.2 | 60.6 | 53.0 | . 60 | . 72 | . 68 | . 50 | .35 | .41 |
| LoRA-Rite$_{r=4}$ | 43.1 | 61.4 | 53.2 | **. 62** | . 73 | . 68 | . 49 | .35 | .41 |
| AdamS$_{r=4}$ | 43.0 | **62.7** | 53.9 | **. 62** | **. 74** | **. 69** | . 49 | .34 | .41 |
| LRACS$_{r=4}$ | **44.0** | 62.5 | **54.0** | **. 62** | **. 74** | **. 69** | **. 48** | **.33** | **.40** |

Table 9: Results of optimizers on GPT-2 medium model in WebNLG challenge. The results show that our proposed method outperforms current methods across BLEU, chrf++ and TER metrics.

approaches: ScaledAdam uses the identity matrix in place of a matrix inverse, while LoRA-Rite applies the Moore–Penrose inverse and uses $nan\_to\_num$ to handle extremely small values that arise during computation.

**Biased Initialization.** As a variant of zero initialization, is a possible way which applies nonzero initializations both to $M_0$ and $N_0$. That is,

$$[\boldsymbol{M}_0]_{ij} \sim \mathcal{N}\left(0, \alpha^2\right) \quad [\boldsymbol{N}_0]_{ij} \sim \mathcal{N}\left(0, \beta^2\right), \quad \alpha, \beta > 0. \text{ (biased-init)}$$

In this setting, we need to subtract the initial weight matrix $\delta W = \boldsymbol{M}_0 \boldsymbol{N}_0^{\mathrm{T}}$ from the pre-trained weights $W$ to ensure the starting point remains at 0. Although this method works well in our experiments, it requires modifications to the pipeline to support the initialization. Thus we give it up.

Besides the above two starts, we propose using a robust start during training and introducing $robust\_steps$ as a hyperparameter. Before the training step reaches $robust\_steps$, we replace our EFL method with AdamW; after reaching $robust\_steps$, we switch back to our EFL method. The advantage is that we can avoid convergence issues near zero without changing the initialization scheme. In our experiments, $robust\_steps$ is set to 100.

**Spectral Initializaiton.** Denoting one-step gradient of full fine-tuning as $G^\natural$, (Zhang et al., 2025) compute the singular value decomposition (SVD) of $\boldsymbol{G}^\natural = \widetilde{\boldsymbol{U}}_{\boldsymbol{G}^\natural} \widetilde{\boldsymbol{S}}_{\boldsymbol{G}^\natural} \widetilde{\boldsymbol{V}}_{\boldsymbol{G}^\natural}^\top$. Then they apply a nonzero

| Method | learning rate | CLIP ↑ | Method | learning rate | CLIP ↑ |
|---|---|---|---|---|---|
| | $1e-4$ | 32.11 | | $1e-3$ | 31.19 |
| AdamS | $2e-4$ | 32.20 | LRACS | $2e-3$ | 31.42 |
| | $5e-4$ | 31.95 | | $5e-3$ | 31.08 |
| | batch size | CLIP ↑ | | batch size | CLIP ↑ |
| | 1 | 32.13 | | 1 | 30.86 |
| AdamS | 2 | 32.20 | LRACS | 2 | 31.42 |
| | 4 | 32.30 | | 4 | 32.16 |

Table 10: Results of optimizers on Mix-of-show model in sensitivity analysis. The results show that our proposed methods are robust to hyperparameter changes

| Method | learning rate | WebNLG | | | | | | | | |
|---|---|---|---|---|---|---|---|---|---|---|
| | | BLEU ↑ | | | chrf++ ↑ | | | TER ↓ | | |
| | | U | S | A | U | S | A | U | S | A |
| | $1e^{-4}$ | 43.40 | 61.72 | 53.62 | . 62 | **. 74** | . 68 | . 49 | .34 | **.41** |
| AdamS | $2e^{-4}$ | 42.95 | **62.73** | 53.92 | . 62 | **. 74** | **. 69** | . 49 | .34 | **.41** |
| | $5e^{-4}$ | 43.28 | 62.38 | 53.42 | . 62 | **. 74** | . 68 | . 49 | .34 | **.41** |
| | $1e^{-3}$ | 43.51 | 61.99 | 53.75 | **. 63** | **. 74** | **. 69** | . 49 | .34 | **.41** |
| LRACS | $2e^{-3}$ | **44.02** | 62.42 | **53.94** | . 62 | **. 74** | **. 69** | **. 48** | **.33** | **.41** |
| | $5e^{-3}$ | 43.29 | 61.46 | 53.39 | . 62 | **. 74** | **. 69** | . 49 | .34 | **.41** |

Table 11: Results of optimizers on GPT-2 medium model in sensitivity analysis. The results show that our proposed methods are robust to hyperparameter changes

initialization strategy, termed spectral initialization.

$$
\begin{aligned}
\boldsymbol{M}_0 &= \sqrt{\gamma} \left[\widetilde{\boldsymbol{U}}_{\boldsymbol{G}^\natural}\right]_{[:,1:r]} \left[\widetilde{\boldsymbol{S}}_{\boldsymbol{G}^\natural}^{1/2}\right]_{[1:r]}, \\
\boldsymbol{N}_0 &= \sqrt{\gamma} \left[\widetilde{\boldsymbol{S}}_{\boldsymbol{G}^\natural}^{1/2}\right]_{[1:r]} \left[\widetilde{\boldsymbol{V}}_{\boldsymbol{G}^\natural}\right]_{[:,1:r]}^\top,
\end{aligned}
\qquad \text{(Spectral-init)}
$$

This initialization can be quite beneficial for LoRA-S because the starting point is no longer zero. However, it requires access to G, which means registering hooks during backpropagation and ensuring they are called successfully. Such pipeline modifications are overly complex, are not adopted by most LoRA methods, and cannot be achieved solely by changing the optimizer. For ease of use, we eventually did not adopt this approach. However, this initialization is very favorable for LoRA-S, as it eliminates the need for the $robust\_steps$ hyperparameter.

### A.17 LOW-MEMORY VARIANTS OF ADAMS AND LRACS

A low-memory variant of AdamS, inspired by the structure of the tangent space of the popular manifold $\mathcal{M}(r, m \times n)$. Although the descent vector $\dot{X}_{MN^\mathsf{T}}$ occupies $n \times m$ space to store, when viewed in manifold coordinates it only has $d = mr + nr - r^2$ dimensions. From Equation (2), we can see that the projected gradient can be decomposed into three mutually orthogonal components $S$, $M_p$ and $N_p$:

$$
M_p \leftarrow \nabla M R_N^\dagger - UU^\mathsf{T}\nabla M R_N^\dagger \tag{23}
$$
$$
N_p^\mathsf{T} \leftarrow R_M^\dagger \nabla N - R_M^\dagger \nabla N V V^\mathsf{T}
$$
$$
S \leftarrow U^\mathsf{T}\dot{X}_{MN^\mathsf{T}}V \tag{24}
$$

By applying the first momentum to these components, we can then recover the overall first moment together with the current coordinates $U, V$ solved from QR decomposition $M = UR_M, N = VR_N$,

which greatly reduced the storage required for the first-order momentum. That is,

$$\text{mom}(M_p) \leftarrow \beta_1 \text{mom}(M_p) + (1 - \beta_1) M_p$$
$$\text{mom}(N_p^{\mathrm{T}}) \leftarrow \beta_1 \text{mom}(N_p^{\mathrm{T}}) + (1 - \beta_1) N_p^{\mathrm{T}}$$
$$\text{mom}(S) \leftarrow \beta_1 \text{mom}(S) + (1 - \beta_1) S$$
$$m_t \leftarrow U\text{mom}(S)V^{\mathrm{T}} + \text{mom}(M_p)V^{\mathrm{T}} + U\text{mom}(N_p^{\mathrm{T}}) \qquad (25)$$

The memory used by $\text{mom}(M_p), \text{mom}(N_p^{\mathrm{T}})$ and $\text{mom}(S)$ is only $mr + nr + r^2$, which is greatly reduced comparing with $n \times m$. However, this technique cannot solve the storage problem of the second moment, because there is no mutually orthogonal coordinate structure within the second moment. Therefore, it can only halve the memory usage, and the overall stroge remains $\mathcal{O}(mn)$. Finally, we present the complete low-memory variant algorithm:

---

**Algorithm 4 AdamS (lm)**

---

1: **Input:** learning rate $\lambda$, first momentum decay $\beta_1$, second momentum decay $\beta_2$, optimization steps $T$. **Init** with $m_0 = 0; v_0 = 0$
2: **for** $t = 1 \ldots T$ **do**
3:     Obtain the descent vector $\dot{X}_{M_t N_t^{\mathrm{T}}}$ from (12);
4:     Obtain the decomposed components $M_p, N_p^{\mathrm{T}}, S$ from (23 - 24);
5:     Calculate first momentum $m_t$ from (25);
6:     Calculate second momentum $v_t \leftarrow \beta_2 v_{t-1} + (1 - \beta_2) \dot{X}^2_{M_t N_t^{\mathrm{T}}}$;
7:     Calculate preconditioned vector $\tilde{X}_{M_t N_t^{\mathrm{T}}} \leftarrow m_t / \sqrt{v_t}$;
8:     Solve $K_t$ from the Sylvester equation (9) to lift $\tilde{X}_{M_t N_t^{\mathrm{T}}}$;
9:     Compute the lift vector $\dot{X}_{\uparrow \mathrm{M}(M_t, N_t)} \leftarrow \left( \tilde{X}_{M_t N_t^{\mathrm{T}}} N_t - M_t N_t^{\mathrm{T}} N_t K_t \right) \left( N_t^{\mathrm{T}} N_t \right)^{-1}$;
10:     Compute the lift vector $\dot{X}_{\uparrow \mathrm{N}(M_t, N_t)} \leftarrow \left( \tilde{X}^{\mathrm{T}}_{M_t N_t^{\mathrm{T}}} M_t - N_t M_t^{\mathrm{T}} M_t K_t^{\mathrm{T}} \right) \left( M_t^{\mathrm{T}} M_t \right)^{-1}$;
11:     Update model parameters for $(M_{t+1}, N_{t+1}) \leftarrow (M_t + \dot{X}_{\uparrow \mathrm{M}(M_t, N_t)}, N_t + \dot{X}_{\uparrow \mathrm{N}(M_t, N_t)})$;
12: **end for**

---

In our experiments, the low-memory version of the algorithm shows no noticeable change in performance or training speed, but significantly reduces memory usage.

For LRACS in Algorithm 3, the projected gradient $G$ (12) is by default stored as an $n \times m$ matrix, but this state is transient and can be released after use, so it does not significantly increase runtime memory in our experiment. Memory usage can be further reduced by storing $G$ in its decomposed form $G = M_p V^{\mathrm{T}} + U N_p^{\mathrm{T}} + U S V^{\mathrm{T}}$, which requires only $\mathcal{O}(mr + nr)$ space. The computation of $s_t$ and $q_t$ from (14) can then be performed directly from the decomposed form in $\mathcal{O}(mr^2 + nr^2)$ time.

## A.18 RELATED OPTIMIZERS

In this section, we list several first-order algorithms commonly used as baselines in both pre-training and LoRA training. We first start with pre-training optimizers and show how to fit them into our general scheme step by step.

### A.18.1 SGD

Stochastic gradient descent (SGD) is the simplest first-order method leveraging gradient information, which is wildly used in almost all neural networks. and replace Sylvester matrix $K$ with weight decay matrix $\epsilon_{decay} I$. Thus it does not admit efficient feature learning. However, lifting SGD is simple. Firstly, assign the descent vector as the gradient $\nabla_X^{(e)} \mathcal{L}(M_t, N_t)$ then calculate Sylvester matrix $K$. That is,

$$\dot{X}_{M_t N_t^{\mathrm{T}}} = -\nabla_X^{(e)} \mathcal{L}(M_t, N_t)$$

Because this method does not use preconditioning, its training performance is still not as good as other EFL methods.

### A.18.2 ADAMW

AdamW was first proposed in (Loshchilov & Hutter, 2017). By applying diagonal preconditioning, it is now one of thet popular and efficient optimizers in neural networks. It is also the most common baseline in the LoRA framework. As discussed in (Yen et al., 2025), AdamW did not achieve efficient feature learning. We propose a lifting version of Adam, Adam-Sylvester (AdamS) 2, to ensure that. The details are already explained in section (4.2).

### A.18.3 SHAMPOO

Shampoo is proposed by (Gupta et al., 2018) for stochastic tensor optimization. It applies Kronecker product approximation to FIM through power iteration algorithm. Preconditioner of Shampoo can be derived by minimizing an upper bound of $\min_{\tilde{F}} \|\tilde{F} - F\|_F^2$ with structural assumption:

$$\mathcal{H} = \left\{ R_n^{\frac{1}{2}} \otimes L_m^{\frac{1}{2}}; \ R_n \in \mathbb{R}^{n \times n}, \ L_m \in \mathbb{R}^{m \times m} \right\}$$

where $R_n$ and $L_m$ are symmetric positive definite (SPD) matrices. The problem of applying shampoo to transformer-based networks is that the computation cost $O(m^3 + n^3)$ and memory cost $mn + m^2 + n^2$ are too high. Here we provide a lifting version of Shampoo, however, its feasibility still needs to be determined and details need to be refined.

$$\dot{X}_{M_t N_t^{\mathrm{T}}} \leftarrow \delta X = -L_t^{-1/4} \nabla_X f_{\mathcal{L}}(M_t N_t^{\mathrm{T}}) R_t^{-1/4} \quad \text{(Lift Shampoo)}$$

### A.18.4 RACS

Row and Column Scaled SGD (RACS) is proposed by (Gong et al., 2025) as a LLM specialized efficient optimizer. By applying diagonal Kronecker product structure preconditioning, it performs strongly on pre-training the 1B LLaMA. RACS is highly memory efficient since it only needs the storage of two diagonal matrices $S$ and $Q$ and a scalar for the limiter, consuming $\mathcal{O}(m + n + 1)$ memory. In this paper, we lifted RACS to the LoRA framework as LRACS, see section (4.3), and observed very good performance on the GPT-2 LoRA fine-tuning. We hope this phenomenon can inspire the strategy for further optimizers design.

**Now we begin to analyze optimizers degined specific to LoRA framework.**

### A.18.5 SCALED GD (RGD)

Scaled GD is proposed by (Tong et al., 2021). Since it applies a Riemannian metric $g_{(M,N)}$ on top of SGD:

$$g_{(M,N)}((\dot{M}, \dot{N}), (\dot{M}, \dot{N})) = \mathrm{trace}(N^{\mathrm{T}} N \dot{M}^{\mathrm{T}} \dot{M} + M^{\mathrm{T}} M \dot{N}^{\mathrm{T}} \dot{N})$$

The metric is different the metric in this paper (7) and this method also guaranties EFL. However, because it does not use preconditioning, its training performance is fairly poor. From the perspective of our scheme, Scaled GD is equivalent to the following setting:

$$\dot{X}_{M_t N_t^{\mathrm{T}}} = -\nabla_X^{(e)} \mathcal{L}(M_t, N_t)$$
$$K_t = \epsilon_{decay} I$$

We use this method as a baseline for EFL methods.

### A.18.6 QUOTIENT GD

Quotient space GD on fixed-rank manifold was proposed by (Mishra et al., 2014). To solve low-rank matrix completion problem, they proposed two quotient space gradient descent methods, $GH^{\mathrm{T}}$ gradient descent and MMMF gradient descent with respect to the metric in this paper (7) and Euclidean metric. These Riemannian methods guarantee transformation invariance. However, since the usage of Euclidean metric will lead to non-Riemannian quotient manifold which conflicts to invariance condition (4) in Definition (2), MMMF failed to meet our requirement for efficient feature learning. Thus we conduct our experiments with $GH^{\mathrm{T}}$ gradient descent as quotient GD method. Similar with Scaled GD, quotient GD does not use preconditioning, its training performance is fairly poor. We use this method as a baseline for EFL methods.

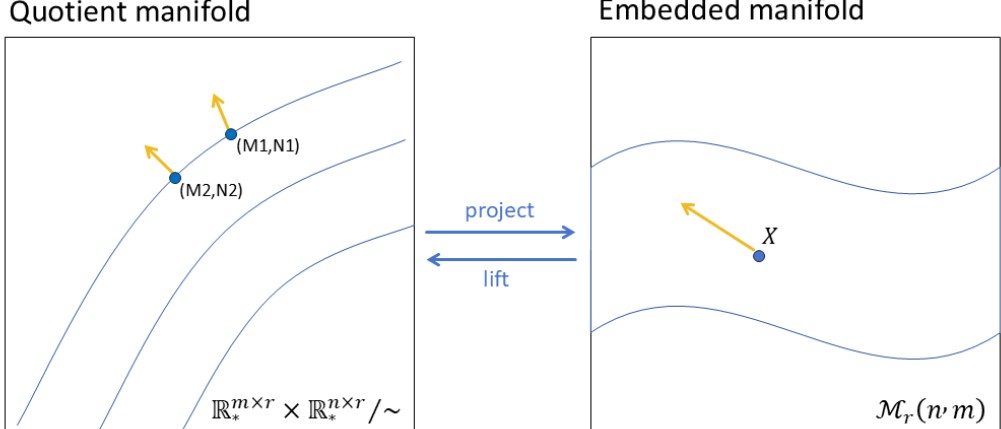

Figure 5: Quotient manifold VS embedded manifold. When $(M1, N1)$ and $(M2, N2)$ represent same point $X \in \mathcal{M}_r(n, m)$, the update vectors (in yellow) at $(M1, N1)$ and $(M2, N2)$ should keep same magnitude to guarantee efficient feature learning.

### A.18.7 SCALED ADAMW

Scaled AdamW is proposed by (Zhang & Pilanci, 2024). It combines the same Riemannian metric with Scaled GD (A.18.5) and diagonal preconditioning from Adam (Kingma & Ba, 2014) in its design. Since they ignore the horizontal lift theory and apply preconditioning to the lifting space, according to (Yen et al., 2025), they also failed to achieve transformation invariance. As stated in AdamS (4.2), the combination of diagonal preconditioning and Riemannian metric should obey our general scheme to guarantee efficient feature learning.

### A.18.8 RIEMANNIAN ADAPTIVE METHODS

Since the LoRA framework is naturally related with Riemannian manifold $\mathcal{M}_r(n, m)$, there is possibility to apply Riemannian adaptive methods (Becigneul & Ganea, 2019) to LoRA. These methods mainly carry the optimizers from Euclidean space, i.e. AMSgrad (Tran et al., 2019), Adagrad (Ward et al., 2020) and Adam, to tangent space assigned with a specific Riemannian metric, i.e. RAMSgrad, RAdagrad and RAdam. These methods do not discuss about efficient feature learning since they only design algorithms on the matrix tangent space $T_{MN^\mathrm{T}}$. However, efficient feature learning require methods to care more about behaviors of update factors on the quotient space $\mathbb{R}_*^{m \times r} \times \mathbb{R}_*^{n \times r} / \sim$. The differences can be seen in Fig (5). All of them work in the right space, but not in the left one.

Besides, our scheme does not use a Riemannian gradient and aiming to remove unstable parts of Riemannian adaptive methods which may lead to divergence, i.e. SVD on manifold, and only focuses on memory and time efficient updates on transformer-based networks, which leads to better performance than any previous Riemannian methods.

### A.18.9 LORA-RITE

LoRA-Rite (Yen et al., 2025) is a method that guarantees EFL, and it is also the first transformation-invariant method with preconditioning. Our work is inspired a lot by LoRA-Rite; in this paper, we aim to extend the favorable properties of LoRA-Rite to a broader family of methods. Under the horizontal lift theory (section 2.2), LoRA-Rite can be viewed as an lift version of Adagrad under a Riemannian metric (6) that is different from ours (7), thereby guaranteeing EFL and inheriting Adagrad-like preconditioning effects. In their metric $\langle \cdot, \cdot \rangle_0$, three mutually orthogonal components $M_G$, $U_G$ and $V_G$ in equation (26) of the descent vector $G = \dot{X}_{M_t N_t^\mathrm{T}}$ are applied with same inner

products, as shown in (27).

$$G = UM_GV^{\mathrm{T}} + U_GV^{\mathrm{T}} + UV_G^{\mathrm{T}} \tag{26}$$

$$\langle G, G \rangle_0 = \mathrm{tr}(N^{\mathrm{T}}N\dot{M}^{\mathrm{T}}\dot{M} + M^{\mathrm{T}}M\dot{N}^{\mathrm{T}}\dot{N})$$

$$= C\,\mathrm{tr}((G^{\mathrm{T}}G)^{-1}VM_G^{\mathrm{T}}M_GV^{\mathrm{T}} + (G^{\mathrm{T}}G)^{-1}VU_G^{\mathrm{T}}U_GV^{\mathrm{T}} + (G^{\mathrm{T}}G)^{-1}V_GV_G^{\mathrm{T}}), \tag{27}$$

where $C$ is a constant. In comparison, our scheme $\langle G, G \rangle_1$ satisfies:

$$\langle G, G \rangle_1 = \mathrm{tr}((M^{\mathrm{T}}M)^{-1}\dot{M}^{\mathrm{T}}\dot{M} + (N^{\mathrm{T}}N)^{-1}\dot{N}^{\mathrm{T}}\dot{N})$$

Then we dig into $(M^{\mathrm{T}}M)^{-1}\dot{M}^{\mathrm{T}}\dot{M}$ and use lifting scheme (1) to substitute $\dot{M}$ and $\dot{N}$:

$$\mathrm{tr}((M^{\mathrm{T}}M)^{-1}\dot{M}^{\mathrm{T}}\dot{M}) = \mathrm{tr}((M^{\mathrm{T}}M)^{-1}(N^{\mathrm{T}}N)^{-1}N^{\mathrm{T}}G^{\mathrm{T}}GN(N^{\mathrm{T}}N)^{-1}) \tag{28}$$

$$+ \mathrm{tr}((M^{\mathrm{T}}M)^{-1}(N^{\mathrm{T}}N)^{-1}K^{\mathrm{T}}N^{\mathrm{T}}NM^{\mathrm{T}}MN^{\mathrm{T}}NK(N^{\mathrm{T}}N)^{-1}) \tag{29}$$

$$- 2\,\mathrm{tr}((M^{\mathrm{T}}M)^{-1}(N^{\mathrm{T}}N)^{-1}K^{\mathrm{T}}N^{\mathrm{T}}NM^{\mathrm{T}}GN(N^{\mathrm{T}}N)^{-1}) \tag{30}$$

Equation (28) indicates the same inner product towards $M_G$, $U_G$ and $V_G$. However, (29) and (30) do not. To see this, we need to apply Sylvester equation (9):

$$M^{\mathrm{T}}GN = R_M^{\mathrm{T}}U^{\mathrm{T}}(UM_GV^{\mathrm{T}} + U_GV^{\mathrm{T}} + UV_G^{\mathrm{T}})VR_N$$

$$= R_M^{\mathrm{T}}M_GR_N$$

$$= M^{\mathrm{T}}MN^{\mathrm{T}}NK + KM^{\mathrm{T}}MN^{\mathrm{T}}N$$

which means a one-to-one relationship between $K$ and $M_G$ independent of $U_G$ and $V_G$. Thus, (29) and (30) apply inner product to $M_G$ only, which leads to different scaled inner products to three mutually orthogonal components $M_G$, $U_G$ and $V_G$ overall. Similar phenomenon also appears in $(N^{\mathrm{T}}N)^{-1}\dot{N}^{\mathrm{T}}\dot{N}$. Different Riemannian metrics also lead to different performances in experiments, and the effects of these Riemannian metrics still need further study.

## A.19 GENERATION RESULTS OF MIX-OF-SHOW MODEL

| Methods | Fig.1 | Fig.2 | Fig.3 | Fig.4 | Fig.5 |
|---|---|---|---|---|---|

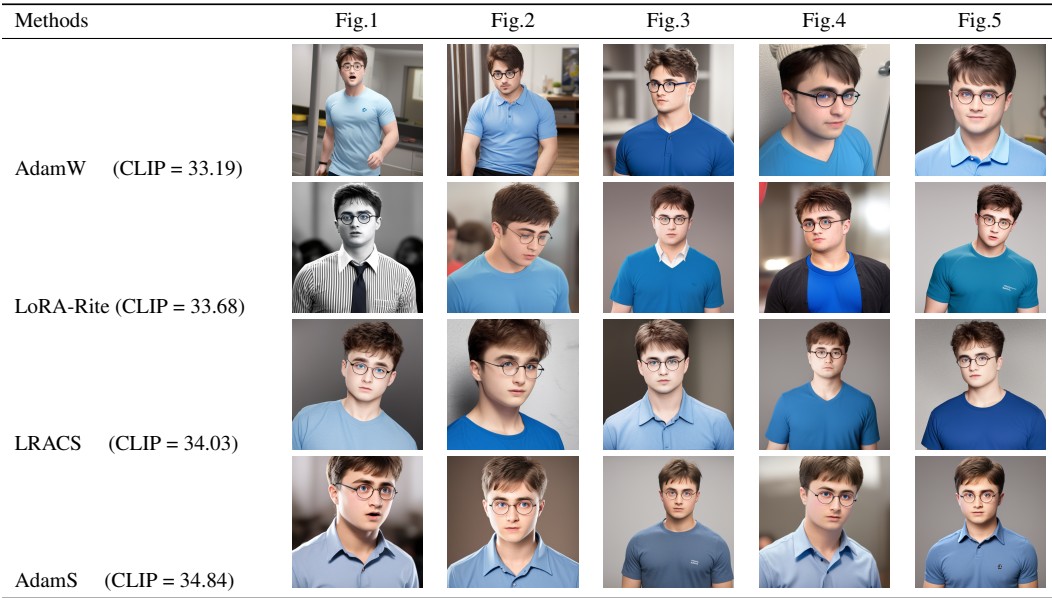

| | | | | | |
|---|---|---|---|---|---|
| AdamW (CLIP = 33.19) | | | | | |
| LoRA-Rite (CLIP = 33.68) | | | | | |
| LRACS (CLIP = 34.03) | | | | | |
| AdamS (CLIP = 34.84) | | | | | |

Figure 6: A batch of generation results visualization of Mix-of-show model with average CLIP score. The prompt is "a $< V_{potter} >$ in blue shirt".

