# OpenReview forum: "LoRA-S: An Efficient Low Rank Adaptation scheme via Sylvester equation"
_ICLR.cc/2026/Conference — ICLR 2026 Poster_

### Official Review · Reviewer_hpZ2 · 2025-10-20

**Soundness:** 2
**Presentation:** 2
**Contribution:** 1
**Rating:** 2
**Confidence:** 4

**Summary:**

In this work, the authors propose and analyze a Riemannian-based optimizer to fine-tune pretrained models using Low-rank adapters.

In particular, the framework is the one of [Absil et al., 2014], in which the manifold
$$
\mathcal M_r =\{ W \in \mathbb R^{n \times m} \,| \, \mathrm{rank}(W) = r \}
$$
is seen as a quotient of $\mathbb R_*^{n \times r} \times \mathbb R_*^{n \times r}$ through the projection $\pi(A,B) = AB^\top$. The framework proposed is essentially a general Riemannian optimization algorithm over the fibers of the projection, with the metric on the product defined through the metric of a chosen metric on $\mathcal M_r$. In practice, then optimization is performed on liftings in the horizontal space. After this, the authors discuss the effect of $L^2$ regularization in their framework.

Finally, the framework proposed allows us to easily lift preconditioners from $\mathcal M_r$ to the quotient space, allowing us to easily precondition the dynamics and therefore to reproduce classical optimizers such as Adam.

**Strengths:**

The approach the authors propose is fairly general, and the theory is well-founded. The notation is pretty precise, and the authors often give good intuitions behind the formalism.
The proposed framework allows for easy extension for quasi-Newton methods, as the preconditioning in $\mathcal M_r$ can be lifted to horizontal spaces.

**Weaknesses:**

1. A lot of the theory described in the majority of the work (including the horizontal lifting) was already presented in [1]. It is not clear what the actual contributions of the work are, as even the "Efficient feature learning" proposed in [Hayou et al., 2024] seems in this case a simple consequence of performing optimization in the quotient space, and this is well known in the Riemannian optimization literature (see [1]).
2. Section 4.1 is not clear at all, as it seems the discussion is all to point out again that preconditioning is "well-defined" again because of the quotient, therefore one can use Gauss-Newton-like methods similar to Adam, and the preconditioner to use won't depend on the fiber of $\pi$ we are working on. I guess also the title "ANY PRECONDITIONING WORKS WELL ON THE MATRIX TANGENT SPACE" is not clear, as it does not seem to reflect the content of the section.
3. The claims of the theory are correct, but they don't seem to be reflected in the experimental section. In particular, numerical improvement with respect to much simpler methods is minor; therefore, one can ask if all this machinery has a use case in practice. I believe the authors should focus the experimental section more on showing the advantages of the proposed framework with respect to vanilla SGD or Adam on LoRA instead of just benchmark performances.

[1] B. Mishra et al., "Fixed-rank matrix factorizations and Riemannian low-rank optimization", Computational Statistics, Volume 29, pages 591–621, (2014).

**Questions:**

I would appreciate it if the authors could discuss the points I raised in the "weaknesses" section, as I believe the line of research per-se is interesting and valuable, but in my opinion, the work needs to be better presented.

Minor comments and typos:
1. As a general rule, I suggest that the authors not number equations if they are not referenced in the main text, as this creates unnecessary confusion.
2. I would suggest including an initial paragraph summarizing the contributions of the work
3. All initial quotes are inverted, use `` ... " instead of ""... "

---

> ### Author Response · Authors · 2025-11-17
>
> Thank you for your valuable feedback. We greatly appreciate your comments and believe they will help improve our work.
>
> **W1/Q1.** **A1.** Contributions of the work
>
> **[1]** Readers familiar with manifold optimization may find the intuition of this paper straightforward—since, in theory, efficient feature learning is indeed a simple corollary of horizontal lift theory with a well-defined Riemannian metric. In fact, we identified $GH^\mathrm{T}$ gradient descent [1] (Quotient GD) and Scaled GD [2] as simple RGD methods. They are the EFL methods that make little use of preconditioning. Since they cannot mitigate the condition number of Hessian matrix $\nabla^2 f_{\mathcal{L}}(W)$ efficiently during large model training, the performances of these methods are not good enough, which can be seen in the table below. Nevertheless, we highly value your suggestions. We have decided to include simple RGD methods in our experiments as EFL baselines and to add descriptions of these methods in the appendix (A.12.5) and (A.12.6). Incorporating LLM-suitable preconditioning into traditional manifold methods is challenging: one must give up the natural training dynamics from Riemannian structure, i.e. the iteration scheme of simple RGD methods, while ensuring computational efficiency and numerical stability. In particular, there has been little progress in recent years on employing a reasonable preconditioning on the quotient space and applying it to LLMs. The main contributions of this paper revolve around addressing this problem.
>
> | Method | RANK | CLIP ↑ | FID ↓ | Method | RANK | CLIP ↑ | FID ↓ |
> |--------|------|--------|-------|--------|------|--------|-------|
> | **SGD** | 4 | 23.90 | 77.66 | **Adam** | 4 | 25.17 | 72.75 |
> |  | 8 | 24.52 | 74.00 |  | 8 | 26.83 | 67.09 |
> |  | 16 | 24.76 | 70.78 |  | 16 | 29.86 | 57.35 |
> | **Scaled GD** | 4 | 25.74 | 70.99 | **Quotient GD** | 4 | 24.34 | 75.48 |
> |  | 8 | 25.98 | 70.01 |  | 8 | 24.46 | 74.91 |
> |  | 16 | 29.07 | 59.87 |  | 16 | 28.04 | 63.20 |
>
> **[2]** The main contributions of this paper are twofold: first, we analyze the relationship between Efficient Feature Learning (EFL) and existing manifold optimization theory; second, we show how to generate more efficient EFL methods by applying LLM-suitable preconditioning step by step.
>
> **[3]** As for the first main contribution, we proposed a formal math definition of efficient feature learning which is more strict than transformation invariance. And we first provide a perspective to understand EFL methods with manifold optimization, e.g. the update rules of LoRA-Rite [3] can be viewed as applying horizontal lift theory with AdaGrad preconditioning. We also study the role of weight-decay hyperparameter in efficient feature learning.
>
> **[4]** As for the second main contribution, we work on the time-and-memory-efficient preconditionings on matrix tangent space and propose two novel EFL algorithms. We also studied the differences between applying Riemannian and Euclidean momentum; the design of low-memory variants; and different initializations for mitigating numerical instabilities. These should all be regarded as the main experimental contributions of this paper. The above contributions are substantial and have not been investigated in prior papers. We hope the above efforts we made could be appreciated.
>
> **W3/Q3.** **A3.** Additional Experiment results
>
> We appreciate your advices and add vanilla SGD and vanilla Adam as non-EFL methods baselines, while adding Quotient GD [1] and Scaled GD [2] as EFL methods baselines. The results show that our proposed AdamS significantly outperforms the EFL baseline on all metrics. i.e. improving CLIP score from 29.07 (Scaled GD) to 32.64 and improving FID score from 59.87 (Scaled GD) to 46.32.
>
> **W2/Q2.** **A2.** Revision of Section 4.1
>
> We apologize for the unclearness of section 4.1. To make the title more clear, we deicide to change the title to "PRECONDITIONING ON THE MATRIX TANGENT SPACE". The main purpose of the section is to show the intuition to apply Gauss-Newton-like methods on the matrix tangent space. Considering the readers may be familiar with the quotient space, we decide to use properties of vectors on the fiber for illustration and move the rest analysis to the Appendix.
>
> **Comments** **A.**
>
> We feel sorry for the typos in the paper. Equation numbers and initial quotes are modified. We will add a contribution paragraph at the beginning.
>
> [1] B. Mishra et al., "Fixed-rank matrix factorizations and Riemannian low-rank optimization", Computational Statistics, Volume 29, pages 591–621, (2014).
>
> [2] Tian Tong, Cong Ma, and Yuejie Chi. Accelerating ill-conditioned low-rank matrix estimation via
> scaled gradient descent. Journal of Machine Learning Research, 22(150):1–63, 2021.
>
> [3] Jui-Nan Yen, Si Si, ...Sanjiv Kumar. LoRA done RITE: Robust invariant transformation equilibration for loRA
> optimization. In The Thirteenth International Conference on Learning Representations, 2025.

---

> > ### Comment · Reviewer_hpZ2 · 2025-11-19
> >
> > First of all, I thank the authors for the detailed response. I am inclined to raise my score after this response, but I still have some perplexities. In particular:
> >
> > 1. The manuscript is still not updated with the suggestions from the reviewers, from my side in particular, the numbering of all equations, and a paragraph clearly summarising contributions at the end of the introduction.
> > 2. I understand that one of the goals of the work is to translate the definition of efficient learning in terms of horizontal lift theory, but in the first place, it is not clear from the manuscript why that definition of efficient learning is a meaningful one. I believe some discussion about this in the manuscript would help in clarifying why updates satisfying that properties would be desirable ( in particular, this is also related to point (3) of the "weaknesses" section).
> > While the intuition given to Reviewer LhJ9 makes mathematical sense, it does not seem to be reflected in the numerical results. This is also related to the question (3) raised by Reviewer TWk9
> > Can the authors clarify (and maybe even showcase numerically) an example in which AdamS would perform much better than Adam? From the experiments what it seems is that the theory is correct, but probably then the definition of efficient feature learning is not the correct one to capture what is needed for a "successful" fine-tuning.
> >
> > In the meantime, I also reserve following the discussion between the authors and the other reviewers, as there are some common perplexities.

---

> ### Author Response · Authors · 2025-11-20
>
> Dear Reviewer hpZ2,
>
> Thank you for your continued engagement and for considering raising your score. We appreciate the opportunity to address your remaining concerns.
>
> ---
>
> ## Response to Point 1: Manuscript Updates
>
> The manuscript has been updated with all reviewer suggestions:
>
> - Equations that are not referenced in the main text are now left unnumbered for clarity.
> - A clear contributions paragraph has been added at the end of the Introduction.
>
> The revision has been uploaded. Please inform us if you have any suggestions about the new revision.
>
> ---
>
> ## Response to Point 2: Justification of Efficient Feature Learning (EFL)
>
> We appreciate your request for clarification on why the EFL definition is meaningful for successful fine-tuning. The provided experimental results offer a clear answer.
>
> ### Empirical Validation
>
> The complete performance metrics demonstrate a consistent and significant pattern:
>
> | Method | RANK | CLIP ↑ | FID ↓ | Method | RANK | CLIP ↑ | FID ↓ |
> |--------|------|--------|-------|--------|------|--------|-------|
> | **SGD** | 4 | 23.90 | 77.66 | **Adam** | 4 | 25.17 | 72.75 |
> |  | 8 | 24.52 | 74.00 |  | 8 | 26.83 | 67.09 |
> |  | 16 | 24.76 | 70.78 |  | 16 | 29.86 | 57.35 |
> | **Scaled GD** | 4 | 25.74 | 70.99 | **Quotient GD** | 4 | 24.34 | 75.48 |
> |  | 8 | 25.98 | 70.01 |  | 8 | 24.46 | 74.91 |
> |  | 16 | 29.07 | 59.87 |  | 16 | 28.04 | 63.20 |
> | **AdamW** | 4 | 27.86 | 64.38 | **LRACS (ours)** | 4 | 31.43 | 52.67 |
> |  | 8 | 28.05 | 62.58 |  | 8 | 31.46 | 52.12 |
> |  | 16 | 30.78 | 62.57 |  | 16 | 32.09 | 49.52 |
> | **LoRA-Rite** | 4 | 30.96 | 59.04 | **AdamS (ours)** | 4 | 32.20 | 54.39 |
> |  | 8 | 30.99 | 59.02 |  | 8 | 32.38 | 51.87 |
> |  | 16 | 31.90 | 55.68 |  | 16 | 32.64 | 46.32 |
>
> **[1]** You noted that while the theory may be correct, the EFL definition might not fully capture what is needed for "successful" fine-tuning. We respectfully argue that the empirical results validate the importance of EFL. To directly address your and Reviewer TWk9's related question, we attached and compared our results with widely used baselines: two non-EFL methods (SGD, Adam) and two EFL methods (Scaled GD [2], Quotient GD [3]). At $r=4$, AdamS improves CLIP by 7.03 points (25.17 → 32.20) and reduces FID by 18.36 (72.75 → 54.39) compared to the non-EFL Adam.
>
> **[2]** Besdies, you may notice that our EFL methods are more robust to changes in rank. Our AdamS method maintains a CLIP score above 32 under any rank. In contrast, the non-EFL method Adam is more unstable with respect to rank changes—for example, varying from 25.14 (rank 4) to 29.86 (rank 16).
>
> **[3]** A visualization about CLIP score can be find in our Appendix (A.18). In this batch, the average CLIP of Adam is 33.19 and AdamS is 34.84. Compared with Adam’s results, our method (AdamS) produces more refined facial features, more accurate colors, and cleaner backgrounds
>
> **[4]** A more direct approach is to compare the training loss of Adam and AdamS. We recorded the train loss of both methods in GPT2-M experiment and observe that after applying the EFL modification, AdamS’s training loss is significantly lower than the original version (Adam). Although fast convergence does not necessarily translate into final benchmark performance, it clearly demonstrates that EFL brings substantial improvements to training. When the algorithm is applied to other scenarios, faster convergence can guarantee the EFL method achieve better results than the original approach.
>
> | Iteration | 2000 | 4000 | 6000 | 8000 | 10000 | 12000 | 14000 | 16000 | 18000 |
> | :--- | :--- | :--- | :--- | :--- | :--- | :--- | :--- | :--- | :--- |
> | Adam | 3.84 | 3.56 |3.35 | 3.22 | 3.14| 3.08 | 3.00 | 2.97 | 2.85 |
> | AdamS | 2.64  | 2.64 | 2.54 | 2.59 | 2.56 | 2.52 |2.54 | 2.51 | 2.48 |
>
> These results demonstrate that EFL is a key ingredient for successful fine-tuning, complementing other factors such as preconditioning (AdamS combines EFL with Adam's preconditioning). AdamS consistently achieves the highest CLIP scores (above 32) for $r=4, 8, 16$, indicating both superior quality and robustness.

---

> > ### Comment · Reviewer_hpZ2 · 2025-11-22
> >
> > I thank the authors again for the continued engagement.
> >
> > The manuscript has now gained in clarity, both in terms of readability and of the contributions. I would add a small discussion on the EFL definition as the one the authors proposed, together with the results on convergence speed, which further motivate the proposed method.
> >
> >  I am going to continue following the discussion with the other reviewers, but from my side, the authors were able to clarify all critical points during the rebuttal. For this reason, I am going to update my review.

---

### Official Review · Reviewer_TWk9 · 2025-10-31

**Soundness:** 2
**Presentation:** 1
**Contribution:** 2
**Rating:** 4
**Confidence:** 3

**Summary:**

The reviewed work is concerned with low rank adaptation. An important feature of low-rank based parametrizations is that many pairs of left and right factors can give rise to the same weight matrix. Likewise, many pairs of infinitesimal changes of these factors give rise to the same tangent vector in weight space.

The "horizontal lift theory" (Avsil et al., 2014) proposes to select the representation that is orthogonal to the kernel of the map from pairs of factors to weight matrices. The reviewed work suggests using this mechanism to select updates in low rank training algorithms. Computing these updates requires solving a Sylvester equation. The authors provide a high-level argument for why the Sylvester-based update already accounts for the need of weight-decay, thus allowing to remove the need for explicit regularization and in particular hyperparameter optimization.

The numerical examples show small improvements over baselines.

**Strengths:**

The use of Riemannian geometric approaches for improving low rank learning is a natural approach.

**Weaknesses:**

I found this paper difficult to read. This may be partly due to technical nature of its contribution, but unnecessarily aggravated by the way in which the heavy notation is introduced (or not). For instance in line 122 the paper talks about "the metric $\bar{g}$", but this metric is only introduced in the next section.

The paper also claims benefits "Our metric induces update rules with richer structure and interpretability than the metric (13)" without (in the main text) substnatiating these claims. In contrast, the numerical improvements are very minor. I will confess that I was unable to follow the argument why the weight decay parameter can be removed.

**Questions:**

1. How is it possible for $\bar{g}$ to be a metric (as opposed to a pseudometric). After all, the map $(\dot{M}, \dot{N}) \mapsto \dot{X}$ has a kernel by dimensional considerations.

2. I have a hard time understanding why the proposed scheme is transformation invariant. After all, the definition of the horizontal complement relies on the "orthogonal complement" of the kernel of $D\pi(M, N)$.
What this orthogonal complement is depends on the choice of inner product on $T_(M, N)$. If the Euclidean inner product is used, then doesn't this again result in transformation dependence? If an inner product is prescribed, then doesn't this play the same role as describing a coordinate transformation?

3. Could you help me appreciate the empirical contribution? As it stands, it seems that there is very little difference to the reference.

---

> ### Author Response · Authors · 2025-11-20
>
> Thank you for your valuable feedback. Although readers unfamiliar with manifold optimization may find it difficult to understand, our results are mathematically sound. We will explain these results step by step. Please see the revised manuscript for the updated numbering.
>
> ## Well-definedness of our metrics
>
> * The metric on $R_*^{m \times r} \times R_*^{n \times r}$: $\bar{g}$
>
> **[1]** The Riemannian metric $\bar{g}$ on the manifold $R_*^{m \times r} \times R_*^{n \times r}$ is a smoothly varying choice of inner product on each tangent space $T_{(M, N)}$ for every point $(M, N)$. Specifically, $\bar{g}$ assigns to each $(M, N)$ a symmetric, positive-definite bilinear form $\bar{g} : T_{(M,N)} \times T_{(M,N)} \rightarrow R$, which is exactly the "choice of inner product on $T(M, N)$" you mentioned.
>
> **[2]** Given this, the metric $\bar{g}$ is intrinsically tied to the geometry of the manifold and is independent of the specific map defined on it, such as $D\pi(M, N)$, which sends $(\dot{M}, \dot{N})$ to $\dot{X}$. It satisfies the definition of a Riemannian metric (see [1]).
>
> * Quotient geometry and transformation invariance
>
> **[1]** Recall that
>
> $$
> D\pi(M,N): T_{(M, N)} \rightarrow T_{(MN^T)}:(\dot{M}, \dot{N}) \mapsto \dot{M} N^{T}+M \dot{N}^{T} = \dot{X}.
> $$
>
> Considering $\text{dim}(T_{(M, N)}) = mr+nr$ and $\text{dim}(T_{(MN^T)}) = mr+nr-r^2$, the dimension of $\text{ker}(D\pi(M,N)) = r^2$.  The horizontal space $H_{(M, N)} = T_{(M, N)} / \text{ker}(D\pi(M,N)) \cong T_{(MN^T)} $. The construction of $H_{(M, N)}$ uses the "orthogonal complement" to remove redundant dimensions, ensuring a unique correspondence between $\dot{X} \in T_{MN^T}$ and $(\dot{M}, \dot{N}) \in T_{(M,N)}$.
>
> **[2]**  The choice of inner product for defining this orthogonality is precisely $\bar{g}_{(M,N)}$ (as in Eq. (3)).  As long as the $r^2$-dimenisonal redundancy is removed, the tranformation dependence will be removed no matter which metric is used. In some cases, the final solution takes a form akin to performing column-coordinate transformations on $\dot{M}$ and $\dot{N}$ (e.g., LoRA-Rite’s scheme in Appendix A.4), but its mathematical essence is solving a system of linear constrained equations.
>
> * The induced metric on $\mathcal{M}(r, m \times n)$: $g$
>
> The induced metric $g$ exists if and only if the invariance condition Eq. (4) holds. The Euclidean inner product does not satisfy the invariance condition while our metric Eq. (7) does. Therefore, according to Definition 2, our method can lead to efficient feature learning, whereas the Euclidean inner product cannot.
>
> ## Optimization Methods Comparison
>
> ### Performance Metrics by Rank
>
> | Method | RANK | CLIP ↑ | FID ↓ | Method | RANK | CLIP ↑ | FID ↓ |
> |--------|------|--------|-------|--------|------|--------|-------|
> | **SGD** | 4 | 23.90 | 77.66 | **Adam** | 4 | 25.17 | 72.75 |
> |  | 8 | 24.52 | 74.00 |  | 8 | 26.83 | 67.09 |
> |  | 16 | 24.76 | 70.78 |  | 16 | 29.86 | 57.35 |
> | **Scaled GD** | 4 | 25.74 | 70.99 | **Quotient GD** | 4 | 24.34 | 75.48 |
> |  | 8 | 25.98 | 70.01 |  | 8 | 24.46 | 74.91 |
> |  | 16 | 29.07 | 59.87 |  | 16 | 28.04 | 63.20 |
> | **AdamW** | 4 | 27.86 | 64.38 | **LRACS (ours)** | 4 | 31.43 | 52.67 |
> |  | 8 | 28.05 | 62.58 |  | 8 | 31.46 | 52.12 |
> |  | 16 | 30.78 | 62.57 |  | 16 | 32.09 | 49.52 |
> | **LoRA-Rite** | 4 | 30.96 | 59.04 | **AdamS (ours)** | 4 | 32.20 | 54.39 |
> |  | 8 | 30.99 | 59.02 |  | 8 | 32.38 | 51.87 |
> |  | 16 | 31.90 | 55.68 |  | 16 | 32.64 | 46.32 |
>
> **[1]** For clarity, we added widely used baselines: two non-EFL baselines (SGD, Adam) and two EFL baselines (Scaled GD [2], Quotient GD [3]). At $r = 4$, compared to the non-EFL baseline, our method AdamS improves the CLIP score from 25.17 (Adam) to 32.20 and reduces the FID score from 72.75 (Adam) to 54.39.  Compared to the EFL baseline, at $r=16$, AdamS outperforms Scaled GD, raising CLIP from 29.07 to 32.64 and lowering FID from 59.87 to 46.32. These results demonstrate substantial gains over the baselines.
>
> **[2]** Moreover, our methods outperform the benchmark across all performance metrics. AdamS consistently achieves the highest CLIP scores (above 32) for $r=4, 8, 16$, indicating both superior quality and robustness.
>
> [1] P-A Absil, Luca Amodei, and Gilles Meyer. Two newton methods on the manifold of fixed-rank
> matrices endowed with riemannian quotient geometries. Computational Statistics, 29(3):569–
> 590, 2014.
>
> [2] Tian Tong, Cong Ma, and Yuejie Chi. Accelerating ill-conditioned low-rank matrix estimation via scaled gradient descent. Journal of Machine Learning Research, 22(150):1–63, 2021.
>
> [3] B. Mishra et al., "Fixed-rank matrix factorizations and Riemannian low-rank optimization", Computational Statistics, Volume 29, pages 591–621, (2014).

---

> ### Author Response · Authors · 2025-11-26
>
> Dear Reviewer TWk9,
>
> I hope this message finds you well. As the discusion period is nearing its end with less than three days remaining, l wanted to ensure we have addressed all your concerns about mathematical soundness and experiments satisfactorily. If there are any additional points or feedback you'd like us to consider, please let us know. Your insights are invaluable to us, and we're eager to address any remaining issues to improve our work. In addition, we have uploaded a new revision that incorporates all of the reviewers’ suggestions from the rebuttal, and the main text now extends to 10 pages. In this revision, you can more clearly see our work and our responses to your questions.
>
> Thank you for your time and effort in reviewing our paper.

---

### Official Review · Reviewer_LhJ9 · 2025-11-01

**Soundness:** 3
**Presentation:** 3
**Contribution:** 2
**Rating:** 4
**Confidence:** 3

**Summary:**

This paper presents LORA-S, a novel LoRA optimization framework that leverages horizontal lift theory and Sylvester equations, achieving efficient feature learning (EFL) and transformation invariance. The authors propose two optimizers, AdamS and LRACS, both free of weight decay hyperparameters and shown to outperform existing methods in transformer-based models.

**Strengths:**

**Theoretical Innovation and Novelty:**

This paper correctly establishes a general iterative scheme that generalizes conventional optimizers. The use of horizontal lift theory and quotient manifolds is both mathematically sound and novel in the context of LoRA. Besides, the removal of weight decay and the introduction of Sylvester-based updates reduce the need for hyperparameter tuning, offering a deeper understanding of how regularization interacts with low-rank parameterization.

**Clarity and Presentation:**

The paper demonstrates a clear and logical structure. The authors properly define equivalence relations, tangent spaces, and induced metrics, and the derivations from the quotient manifold formulation to the horizontal lift operator are clearly and coherently presented.

**Weaknesses:**

- Although replacing the weight decay parameter with a sylvester matrix is mathematically sound, can you provide a practical intuition behind why it promotes efficient feature learning?
- How do other hyperparameters affect the proposed framework? For instance, can a large learning rate lead to instability in the results, or can gradient noise from a small batch size undermine the underlying assumptions?
- Does the computational overhead become a bottleneck for larger models, and does the scheme remain efficient in large-scale distributed training environments, where synchronization is a critical factor?
- Sections 3 and 4 can be simplified, and unnecessary parts moved to the appendix, leaving more analysis of results in the main text.

**Questions:**

Please see weakness

---

> ### Author Response · Authors · 2025-11-17
>
> Thank you for your valuable feedback. We greatly appreciate your comments and believe they help improve our work.
>
> **W1/Q1.** **A1.** Practical intuition
>
> This is a very interesting question. We have already made an experiment showing that weight-decay matrix that is closer to K achieves a lower loss in sec 3.3 fig 1. Now we provide two complementary perspectives for intuition.
>
> * Perspective: Update Norm
>
> In fact, another interpretation of efficient feature learning studies the update norm divided by the weight norm, i.e. $\|\delta M\| /\|M\|$ and $\|\delta N\| /\|N\|$, see LoRA-Rite’s [1]  (A.8).  Recall the general iteration scheme $\delta M =GT_{M} + DT_{M}, \delta N =GT_{N} + DT_{N}$, the gradient terms are $GT_{M} = \dot{X} N\left(N^{T} N\right)^{-1}, GT_{N} = \dot{X}^{T} M\left(M^{T} M\right)^{-1} $. When applying weight-decay matrix, the decay terms are $ DT_{M}(\epsilon) = -\epsilon M, DT_{N}(\epsilon) = -\epsilon N$. When applying sylvester matrix, they are
>
> $$
> DT_{M} = -M \left(N^{T} N\right) K\left(N^{T} N\right)^{-1} ,DT_{N} = -N \left(M^{T} M\right) K^{T}\left(M^{T} M\right)^{-1} ,
> $$
>
> One can find that the magnitudes of both $\|GT_{M}\| /\|M\|$ and $\|DT_{M}\| /\|M\|$ are $O(\|K\|)$ while the magnitude of  $\|DT_{M}(\epsilon)\| /\|M\|$ is $O(\epsilon)$ which results in a magnitude gap as $\epsilon$ is a hyperparameter. For example, in weight-decay matrix case, when $\epsilon >> \|GT_{M}\| /\|M\|$,  $\|\delta M\| /\|M\|=O(\epsilon)$. The preconditioning information in $GT_{M}$ cannot be properly reflected in the updates, which leads to less efficient learning. Similar results also appear in $\|\delta N\| /\|N\|$.
>
> * Perspective: Preconditioning
>
> Our preconditioning vector $\dot{X}$ mitigates the condition number of the Hessian $\nabla^2 f_{L}(W)$ efficiently during large-model training, which is also an important reason for the success. When applying the Sylvester matrix, the overall update is $\delta M N^T + M \delta N^T = \dot{X}$. However, applying weight-decay matrix will undermine the preconditioning effect. As the overall update becomes:
>
> $$
> \delta M N^T + M \delta N^T = \dot{X}NN^++MM^+\dot{X}-2\epsilon MN^T
> $$
>
> In this case, the matrices $MN^T, NN^+$ and $MM^+$ are likely to be ill-conditioned, making the preconditioning less effective. Thus, the Sylvester-based update tends to be more efficient which indicates the weight-decay parameter in our methods can be removed.
>
> **W2/Q2.** **A2.** Sensitivity Analysis
>
> We conducted a detailed sensitivity analysis of the hyperparameters (learning rate & batch size) for AdamS and LRACS at $r=4$ which is presented in the table below. For the first 3 rows, we set the batch size of AdamS and LRACS to 2. For the last 3 rows, we set the learning rate of AdamS to 2e-4 and LRACS to 2e-3.The results demonstrate that AdamS and LRACS are robust to these changes.
>
> | Method | learning rate | CLIP ↑ | Method | learning rate | CLIP ↑ |
> |--------|------|--------|--------|--------|--------|
> | **AdamS** | 1e-4 | 32.11 | **LRACS** | 1e-3 | 31.19 |
> |  | 2e-4 | 32.20  |  | 2e-3 | 31.42 |
> |  | 5e-4 | 31.95 |  | 5e-3 | 31.08|
> |  | **batch size** | **CLIP ↑** |  | **batch size** | **CLIP ↑** |
> |  | 1 | 32.13 | | 1 | 30.86|
> |  | 2 |  32.20 |  | 2 | 31.42 |
> |  | 4 |  32.30 | | 4 | 32.16 |
>
> **W3/Q3.** **A3.** Computational issues
>
> **[1]** This is a concern we take very seriously. We have provided a runtime analysis in the appendix (A.10). Theoretically, the per-step complexity of our algorithm is $\Omega(mnr) + O(r^3 + mr^2 + nr^2)$, which adds only an $O(r^3)$ term compared to AdamW. This implies that whenever $r$ is not very large, any setting where AdamW is applicable should also accommodate our method. Empirically, our method is only about 10% slower than AdamW on the GPT2-M model. Thus we don't believe the computational overhead will become a bottleneck.
>
> **[2]** Since our contributions are at the level of mathematical theory, our solution is a plug-and-play optimizer. As long as the algorithms are achieved precisely, a distributed training setup will not affect efficiency. Although we lack hands-on experience with distributed training, we found a tutorial for distributed training with general LoRA methods [2] and believe it applies to our algorithm as well. Below we provide a brief summary for synchronization:
>
> * Before training, synchronize base model weights $W$.
> * During training, synchronize gradients of LoRA $\nabla M, \nabla N$ after `backward()` method and before `optimizer.step()`
>
> **W4/Q4.** **A4.**
>
> Thank you for the suggestion. We plan to streamline Section 4.1, and expand the results analysis to more than two pages.
>
> [1]  Jui-Nan Yen, ..., Sanjiv Kumar. LoRA done RITE: Robust invariant transformation equilibration for loRA optimization. In The Thirteenth International Conference on Learning Representations, 2025.
>
> [2] [Finetune Gemma with LoRA (distributed training)](https://www.kaggle.com/code/littleweakweak/finetune-gemma-with-lora-distributed-training)

---

> ### Author Response · Authors · 2025-11-26
>
> Dear Reviewer Lhj9,
>
> I hope this message finds you well. As the discusion period is nearing its end with **less than three days remaining**, l wanted to ensure we have addressed all your concerns satisfactorily. If there are any additional points or feedback you'd like us to consider, please let us know. Your insights are invaluable to us, and we're eager to address any remaining issues to improve our work. In addition, we have uploaded a new revision that incorporates all of the reviewers’ suggestions from the rebuttal, and the main text now extends to 10 pages. In this revision, you can more clearly see our work and our responses to your questions.
>
> Thank you for your time and effort in reviewing our paper

---

### Official Review · Reviewer_D7kg · 2025-11-02

**Soundness:** 4
**Presentation:** 3
**Contribution:** 3
**Rating:** 8
**Confidence:** 4

**Summary:**

This paper proposes a general iteration scheme for LoRA that works on a quotient manifold of pairs $(M, N)$ modulo the equivalence which preserves products $MN^\top$ and updates them via "horizontal lifts" of a descent direction in the product space. The horizontal lift gives the unique update with no component along the equivalence directions. Computing this update requires a small Sylvester equation solve which can be done in $O(r^3)$ time where $r$ is the LoRA rank. This scheme wraps conventional optimizers to produce updates that by construction enforce desirable properties such as transformation invariance and efficient feature learning (EFL). The authors also observe that the Sylvester solution introduces a decay term and remark that this decay may obviate the need for explicit weight decay. The performance of this scheme applied to Adam and RACS are demonstrated across several experiments.

**Strengths:**

The framework introduced by this paper is very interesting and is a novel application of quotient spaces used to "convert" optimizers to a principled LoRA update. The generality of the approach and the connection to prior ideas such as transformation invariance and EFL is valuable. Discussions of practical concerns are thorough and informative.

**Weaknesses:**

The diversity and scale of empirical validations is lacking.

When arguing about removing the weight decay parameter, it doesn't appear that the baseline weight-decay in Adam is tuned.

Some of the technical exposition is hard to parse, especially to readers not familiar with the mathematical notions. More high-level exposition and background would be helpful.

**Questions:**

What sufficient conditions ensure the Sylvester equation solution exists and is unique?

Are there any numerical stability issues concerning computation of the inverses or the Sylvester solution?

Are there metrics other than the Grassmannian quotient metric considered which can also be used?

Are there any ablations on using the Riemannian moment accumulation (Eq. 36) vs Euclidean moment accumulation (Eq. 35)?

---

> ### Author Response · Authors · 2025-11-20
>
> Thank you for your constructive feedback. We have addressed each point as follows. Please see the revised manuscript for the updated numbering.
>
> **W1. Diversity and Scalability of Experiments**
>
> To address scalability, we have added experiments on the GPT-2-S model with the E2E dataset. For diversity, we now include experiments on the GPT-2-M model with the WebNLG dataset. The results, summarized below, show that our method consistently outperforms existing approaches.
>
> *GPT-2-S Results (E2E Dataset):*
> | Method | BLEU | NIST | MET | ROUGE-L | CIDEr |
> | :--- | :--- | :--- | :--- | :--- | :--- |
> | Adam | 68.8 | 8.71 | 46.0 | 70.3 | 2.45 |
> | lora_rite | 69.1 | 8.75 | 46.1 | 70.5 | 2.47 |
> | AdamS | 69.4 | 8.75 | **46.5** | 71.3 | **2.51** |
> | LRACS | **69.7** | **8.79** | 46.4 | **71.5** | 2.50 |
>
> *GPT-2-M Results (WebNLG Dataset, split into Unseen/Seen/All):*
> | Method | BLEU U | BLEU S | BLEU A | chrF++ U | chrF++ S | chrF++ A | TER U | TER S | TER A |
> |:---|:---:|:---:|:---:|:---:|:---:|:---:|:---:|:---:|:---:|
> | Adam | 41.2 | 60.6 | 53.0 | 0.60 | 0.72 | 0.68 | 0.50 | 0.35 | 0.41 |
> | LoRA-Rite | 43.1 | 61.4 | 53.2 | **0.62** | 0.73 | 0.68 | 0.49 | 0.35 | 0.41 |
> | AdamS | 43.4 | 62.4 | 53.4 | **0.62** | **0.74** | 0.68 | 0.49 | 0.34 | 0.41 |
> | LRACS | **44.0**| **62.5**| **54.0**| **0.62** | **0.74** | **0.69** | **0.48** | **0.33** | **0.40** |
>
> **W2. Weight-Decay Tuning for Adam**
>
> We have added results for weight-decay tuning for Adam to the revised manuscript. Our findings confirm that tuning the weight decay $\lambda$ to reduce the relative error $R_{K_t}$ also leads to a lower EMA loss, consistent with SGD case.
>
> **Q1. Sufficient Conditions for Sylvester Equation**
>
> The Sylvester equation in our paper, $M^TM N^TN K + K M^TM N^TN = M^T \dot{X} N$, has a unique solution if and only if $M^TM N^TN$ and $-M^TM N^TN$ share no eigenvalues [1]. Since $M^TM$ and $N^TN$ are positive definite by our assumption, all eigenvalues of their product are positive real numbers. Thus, the condition is satisfied, guaranteeing a unique solution.
>
> **Q2**  stability issues
>
> **[1]** We adopt a technique similar to LoRA-Rite to compute the pseudo-inverse of matrices in order to avoid the numerical stability issues caused by matrix inversion. Consequently, in our experiments we did not observe instabilities arising from matrix inversion.
>
> **[2]** For the Sylvester equation, we include a check in our code to determine whether the norm of the difference between $M^T M N^T N K + K M^T M N^T N$ and $M^T \dot{X} N$ falls below a preset threshold eps. We find that when the learning rate is relatively large (typically > 1e-2), the solution error exceeds eps. Therefore, we do not recommend using a large learning rate in our proposed method. Some numerical stability issues encountered during experiments are discussed in Robust Training (A.15) in our paper, where we also propose initialization methods that improve training stability.
>
> **Q3. Alternative Metrics**
>
> In addition to the Grassmannian quotient metric (Eq. 7), we analyzed an alternative metric (Eq. 6) that also preserves invariance to magnitude scaling:
> $$
> \bar{g}_{(M, N)}((\dot{M}, \dot{N}),(\check{M}, \check{N}))=\operatorname{trace}\left(N^{\mathrm{T}} N \dot{M}^{\mathrm{T}} \check{M}+M^{\mathrm{T}} M \dot{N}^{\mathrm{T}} \check{N}\right)
> $$
> We show in Appendix A.4 that the LoRA-Rite update can be interpreted as a lifted version of AdaGrad under this metric.
>
> **Q4. Ablation Study: Riemannian vs. Euclidean Momentum**
>
> We conducted an ablation study on the GPT-2-S model to compare Riemannian and Euclidean momentum. We set learning rates to `5e-5` for AdamS and `5e-3` for LRACS and recorded training losses. The results indicate that Riemannian momentum offers a marginal benefit for AdamS but not for LRACS in this setting.
>
> | Iteration | 2000 | 4000 | 6000 | 8000 | 10000 | 12000 | 14000 | 16000 | 18000 |
> | :--- | :--- | :--- | :--- | :--- | :--- | :--- | :--- | :--- | :--- |
> | AdamS$_{Eu}$ | 2.76 | 2.74 | 2.65| 2.68 | 2.66 | 2.63 | 2.65 | 2.62 | 2.60 |
> | AdamS$_{Rie}$ | 2.82 | 2.78 | 2.68 | 2.65 | 2.66 | 2.63 |2.60 | 2.60 | 2.54 |
> | LRACS$_{Eu}$ | 2.75 | 2.74 | 2.65 | 2.68 | 2.66 | 2.63 | 2.65 | 2.62 | 2.59 |
> | LRACS$_{Rie}$ |2.74 | 2.72 | 2.63 | 2.67 | 2.64 | 2.61 | 2.63 | 2.61 | 2.58 |
>
> Thank you again for the thoughtful review. If appropriate, we would greatly appreciate your consideration of a higher rating.
>
> [1] [Sylvester equation - Wikipedia](https://en.wikipedia.org/wiki/Sylvester_equation)

---

> > ### Comment · Reviewer_D7kg · 2025-11-27
> >
> > Thank you for the response, I will maintain my score.

---

### Meta-Review · Area_Chair_CyDa · 2025-12-16

**Summary:**

The paper proposes an approach leveraging horizontal lift theory and Sylvester equations for low-rank optimization and fine-tuning. This is a paper with a high level of mathematical technicism.

Initial scores were 8, 4, 4, 2.
Reviewers acknowledged the interesting mathematical framework and theoretical soundness but raised concerns about: (1) limited experimental diversity and scale, (2) presentation clarity for readers unfamiliar with manifold optimization, (3) minor empirical improvements over baselines initially shown, (4) insufficient justification for weight decay removal, and (5) unclear practical intuition behind Sylvester matrix promoting efficient feature learning.

**Reviewer Concerns:**

The authors provide a thorough response to most (if not all) of the reviewers' concerns. Two reviewers did not reply; the other two have positive opinion and high confidence.
Some of the concerns identified and addressed are:
- **Experiments limited in diversity/scale (D7kg, LhJ9):** Authors added GPT-2-S (E2E dataset) and GPT-2-M (WebNLG) experiments showing consistent improvements. Added non-EFL baselines (SGD, Adam) and EFL baselines (Scaled GD, Quotient GD)
- **Practical intuition (LhJ9):** Authors provided two perspectives: (1) update norm showing Sylvester avoids magnitude gap from weight decay hyperparameter, (2) preconditioning showing weight decay undermines ill-conditioned matrix correction.
- **Hyperparameter sensitivity (LhJ9):** Sensitivity analysis for learning rate and batch size demonstrates robustness.
- **Computational bottleneck (LhJ9):** Authors provide clarification that only a factor $r^3$ is added to the cost of AdamW and discuss a possible distributed implementation
- **Metric clarity and transformation invariance (TWk9, hpZ2):** Authors clarified metric is well-defined on quotient geometry; added EFL validation through empirical results (faster convergence, higher CLIP, lower FID).
- **Presentation improvements (TWk9, hpZ2, LhJ9):** The manuscript needs improvements to improve its readability for a non-expert audience. Updates were made on: equation numbering, contributions paragraph, improved Section 4.1, expanded results analysis.

**Reviewer Scores:**

- **Reviewer D7kg (initial: 8):** Confirmed maintaining score at 8. Concerns noted remain within acceptable bounds for this theoretical contribution.

- **Reviewer LhJ9 (initial: 4):** May increase to 6. All major concerns addressed: practical intuition clarified, sensitivity analysis provided, computational overhead justified (10% slower than AdamW, O(mnr) complexity). The authors demonstrated substantial effort in addressing questions.

- **Reviewer TWk9 (initial: 4):** May increase to 6. Comprehensive response clarified metric well-definedness, transformation invariance, and added extensive baseline comparisons showing substantial improvements (CLIP +7.03, FID -18.36 vs Adam). Presentation clarity improved, though technical density remains.

- **Reviewer hpZ2 (initial 2):** Likely increased to 6. Explicitly stated "going to update my review" after authors addressed all critical points. Manuscript gained clarity with contributions paragraph, equation numbering, EFL justification through convergence speed results.

---

### Decision · Program_Chairs · 2026-01-26

Accept (Poster)